# On the Dynamics & Transferability of Latent Generalization during Memorization

## Abstract

Deep Networks have been known to have extraordinary generalization abilities, via mechanisms that aren't yet well understood. It is also known that upon shuffling labels in the training data to varying degrees, Deep Networks, trained with standard methods, can still achieve perfect or high accuracy on this corrupted training data. This phenomenon is called *memorization*, and typically comes at the cost of poorer generalization to true labels. Recent work has demonstrated, surprisingly, that such networks retain significantly better latent generalization abilities, which can be recovered via simple probes on their layer-wise representations. However, the origin and dynamics over training of this latent generalization is not well understood. Here, we track the training dynamics, empirically, and find that latent generalization abilities largely peak early in training, with model generalization, suggesting a common origin for both. However, while model generalization degrades steeply over training thereafter, latent generalization falls more modestly & plateaus at a higher level over epochs of training. Next, we design a new linear probe, in contrast with the quadratic probe used in prior work, and demonstrate that it has superior generalization performance in comparison to the quadratic probe, in most cases. Importantly, using the linear probe, we devise a way to transfer the latent generalization present in last-layer representations to the model by directly modifying the model weights. This immediately endows such models with improved generalization, i.e. without additional training. Finally, we use the linear probe to design initializations for Deep Networks, which, in many cases, turn out to be memorization-resistant, without using regularization. That is, Deep Networks with such initializations tend to evade memorization of corrupted labels, which is often accompanied by better generalization, when used with standard training methods alone. Our findings provide a more detailed account of the rich dynamics of latent generalization during memorization, and demonstrate the means to leverage this understanding to directly transfer this generalization to the model & design better model-weight initializations in the memorization regime.

## 1 Introduction

Overparameterized Deep Neural Networks have seen widespread deployment in many fields, due to their remarkable generalization abilities. However, we still don't have a clear understanding of the mechanisms underlying their ability to generalize so well to unseen data. It has also been shown (Zhang et al., 2017; 2021) that overparameterized Deep Networks are capable of achieving high or even perfect training accuracy on datasets, wherein a subset of training data have their labels randomly shuffled. Such models typically have poor generalization performance, i.e. poorer accuracy on test data with correct labels – a phenomenon that has been called *memorization*. It is known (Arpit et al., 2017) that during training, models trained with such corrupted datasets exhibit better generalization during the initial phases of training; however generalization progressively deteriorates as training accuracy improves subsequently.

A recent study (Ketha & Ramaswamy, 2025) has shown that while Deep Networks trained on datasets having corrupted labels tend to exhibit poor generalization, their intermediate layer representations retain a surprising degree of latent generalization ability. This ability can be recovered from such trained networks by using a simple probe – Minimum Angle Subspace Classifier (MASC) – that leverages the subspace geometry of the corrupted training dataset representations, to this end.

Their findings suggest that generalizable features are present in the layer-wise representations of such networks, even when the model fails to utilize them sufficiently. However, the origin and evolution of this latent generalization ability during training is not well understood. It has also not been clear, if this latent generalization can be harnessed to directly improve the model's generalization. More generally, it has not been known if the inductive bias manifested by the structure of Deep Networks and standard training methods, is, in principle, sufficient to resist memorization in favor of generalization, e.g. by simply choosing an appropriate model initialization. Indeed, existing techniques that resist memorization in the label noise setting typically either use regularization (e.g. Arpit et al. (2017); Liu et al. (2020)) or altogether different training paradigms (e.g. Jiang et al. (2018); Han et al. (2018)), to this end. Here, we address these questions.

Our main contributions are listed below.

- For models trained on standard datasets with various degrees of label corruption, we characterize the evolution of the latent ability to generalize over training using MASC (Ketha & Ramaswamy, 2025). We find that as the model exhibits a peak in its test accuracy early in training (Arpit et al., 2017), the MASC test accuracy at all layers also tend to largely peak concurrently with that of the model, albeit at different levels. Following this, the evolution of test accuracies between the model & MASC diverge, with the model showing a marked decline in test accuracies over further epochs of training. In contrast, the MASC test accuracies decline more modestly & tend to plateau higher, which manifests in the improved generalization ability at the end of training, as reported in (Ketha & Ramaswamy, 2025).

- We observe that MASC is a non-linear classifier; and in particular, prove that it is a quadratic classifier. This brings up the possibility that the improved generalization performance of the probe (i.e. MASC) is attributable to the effectiveness of the quadratic nature of the probe itself and may not easily be decodable, e.g. by a linear probe. To address this point, we introduce a simple linear alternative – Vector Linear Probe Intermediate-layer Classifier (VeLPIC). Surprisingly, we find that VeLPIC almost always achieves superior latent generalization performance in comparison to MASC, especially for higher corruption degrees. This establishes that latent generalization during memorization is linearly decodable from layerwise representations.

- By leveraging the linear probe (VeLPIC), we devise a way to directly modify the pre-softmax weights of such Deep Networks, that immediately transfers to the model, the latent generalization performance of VeLPIC (as applied to the last layer). Notably, this is without requiring additional training. This demonstrates that latent generalization present in layerwise representations can be transferred directly to enhance model generalization of Deep Network models, in the memorization regime.

- Using the linear probe (VeLPIC), we propose an initialization strategy for Deep Networks. When used with standard training (without any explicit regularization), we find that this initialization steers the models away from memorization and towards improved generalization, in many cases.

Our experimental setup is detailed in the Appendix Section A.

## 2 RELATED WORK

In influential work, (Zhang et al., 2017; 2021) showed that Deep Networks can achieve perfect training accuracy even with randomly shuffled labels, accompanied by poor generalization. In follow-up work, (Arpit et al., 2017) find that in the memorization regime, networks learn simple patterns first during training. Their work provides a detailed account of the the early dynamics of training. More recently, (Ketha & Ramaswamy, 2025) in fact show that in spite of the fall in model generalization later on in training, the layerwise representations of the model retain significant latent generalization ability. (Arpit et al., 2017) also show that regularization can help models resist memorization in the label noise case.

Analyzing intermediate representations in Deep Networks has been previously explored using kernel-PCA (Montavon et al., 2011) and linear classifier probes (Alain & Bengio, 2018). Notably, (Alain & Bengio, 2018) state that they deliberately did not probe Deep Networks in the memorization setting since they thought that such probes would inevitably overfit. On the contrary, (Ketha

& Ramaswamy, 2025) demonstrate that probes on Deep Networks in the memorization setting, can have enhanced generalization. (Stephenson et al., 2021) show evidence suggesting that memorization occurs in the later layers. Li et al. (2020) show that in the memorization regime, there is substantial deviation from initial weights.

Several training paradigms have been proposed to enhance generalization performance when learning from corrupted datasets. For example, MentorNet (Jiang et al., 2018) introduces a framework wherein a mentor network guides the learning process of a student network by guiding the student model to focus on likely clean labels. Likewise, Co-Teaching (Han et al., 2018) trained two peer networks simultaneously, each selecting small-loss examples to update its counterpart. Early-Learning Regularization (ELR) (Liu et al., 2020) augmented the training objective with a regularization term, towards this end.

Saxe et al. (2013) offer theoretical explanations on generalization for deep linear networks and Lampinen & Ganguli (2018) offer theoretical explanations in the memorization regime. Methodologies such as Canonical Correlation Analysis (Raghu et al., 2017; Morcos et al., 2018) and Centered Kernel Alignment (Kornblith et al., 2019) have been used to characterize training dynamics and network similarity. Representational geometry and structural metrics provide further insights into learned representation properties (Chung et al., 2016; Cohen et al., 2020; Sussillo & Abbott, 2009; Farrell et al., 2019; Bakry et al., 2015; Cayco-Gajic & Silver, 2019; Yosinski et al., 2014).

## 3 TRAINING DYNAMICS OF LATENT GENERALIZATION USING MASC

Ketha & Ramaswamy (2025) investigate the organization of class-conditional subspaces using the training data at various layers of Deep Networks.

These subspaces are estimated via Principal Components Analysis (PCA), specifically, ensuring that they pass through the origin. To probe the layerwise geometry without relying on subsequent layers, they propose a new probe – the Minimum Angle Subspace Classifier (MASC). For a given test input, MASC projects the layer output onto each class-specific subspace, and computes the angles between the original and projected vectors, for each subspace. The label predicted by MASC corresponds to the class whose subspace yields the projected vector with the smallest such angle. We provide a detailed summary of the working of MASC in the Appendix Section A.2.

As shown in Ketha & Ramaswamy (2025), for models trained with corrupted labels, there exists at least one layer where MASC exhibits better generalization than the corresponding trained model. However, the origin & evolution of this latent generalization across training isn't well understood.

Here, we empirically study the behavior of latent generalization, as manifested by MASC, during training. MASC testing accuracy during training for MLP trained on MNIST, CNN trained on Fashion-MNIST and AlexNet trained on Tiny ImageNet are shown in Figure 1. Results with 0% and 100% corruption degrees as well as the results for additional models i.e. MLP trained on CIFAR-10, CNN trained on MNIST, CNN trained on CIFAR-10 for various corruption degrees are presented in Figure 5 and Figure 6, respectively in the Appendix Section B.

Our findings indicate the presence of two distinct phases in the training process, separated by the point at which the model achieves peak test accuracy. For various non-zero degrees of corruption, in most cases (except those with 100% degrees of corruption), the MASC test accuracy largely follows the rise in the model's test accuracy up to this peak. However, beyond the peak, while the model's test accuracy declines significantly, the drop in MASC accuracy is less steep and plateaus at a higher level over the epochs.

For non-zero corruption degrees (except those with 100% corruption), in most cases, for MLP, MASC accuracy on later layers performed better than MASC accuracy on early layers, whereas for CNNs MASC accuracy on early layers performed better.

Our results represent progress in clarifying the origin & evolution of latent generalization by MASC, during training. In particular, given that model generalization & latent generalization show a concurrent initial rise, it suggests the possibility of common mechanisms that drive both in the early phases of training. The subsequent divergence between model generalization & latent generalization is an intriguing phenomenon, whose mechanisms merit future investigation.

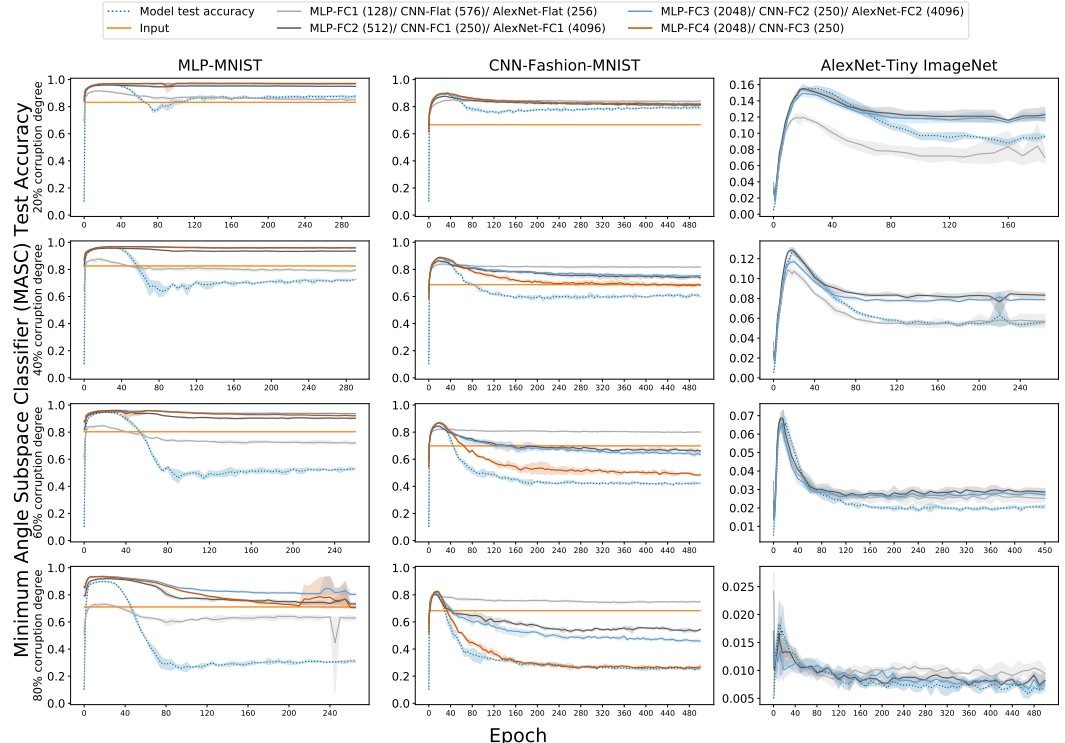

Figure 1: Minimum Angle Subspace Classifier (MASC) test accuracy over epochs of training for multiple models/datasets, where test data is projected onto class-specific subspaces constructed at each epoch from corrupted training data with the indicated label corruption degree. The plots display MASC accuracy across different layers of the network. For reference, the evolution of test accuracy of the corresponding model (blue dotted line) over epochs of training is also shown. FC denotes fully connected layers with $ReLU$ activation, and Flat refers to the flatten layer without $ReLU$.

## 4 NON-LINEARITY OF MASC

Classically (Alain & Bengio, 2018), linear probes have been used to probe layers of Deep Networks. However, (Ketha & Ramaswamy, 2025) do not not use the standard linear probe from (Alain & Bengio, 2018). Below, we prove that MASC (Ketha & Ramaswamy, 2025) is in fact a classifier that is quadratic in the layerwise output of the layer that it is applied to.

**Proposition 1.** *MASC is a quadratic classifier.*

*Proof.* Let $x_l$ denote the output of the layer $l$ of the Deep Network when it is given input $x$. Let $p_1^c, p_2^c, \ldots, p_k^c$ be a basis[1] of the subspace $\mathcal{S}_c$ corresponding to class $c$. Let $x_l^c$ be the projection of $x_l$ on $\mathcal{S}_c$. We have

$$x_l^c = (x_l \cdot p_1^c)\, p_1^c + \ldots + (x_l \cdot p_k^c)\, p_k^c \tag{1}$$

Now, MASC on layer $l$ predicts the label of $x$ as

$$\arg\max_c (x_l \cdot x_l^c) = \arg\max_c ((x_l \cdot p_1^c)^2 + \ldots + (x_l \cdot p_k^c)^2) \tag{2}$$

which is quadratic in $x_l$. This establishes that MASC is a quadratic classifier. □

---
[1]which is typically estimated via PCA, where $k$ is the number of principal components.

# 5 VECTOR LINEAR PROBE INTERMEDIATE-LAYER CLASSIFIER (VELPIC): A NEW LINEAR PROBE

Given that MASC is inherently a non-linear classifier as proved above, a natural question is if its extraordinary ability to decode generalization from hidden representations of memorized networks is a consequence of its non-linearity. Put differently, it raises the question of whether the latent generalization reported in (Ketha & Ramaswamy, 2025) is linearly decodable – with comparable performance – from the layerwise representations of the network.

To build a linear probe analogous to MASC, we sought to retain the same broad idea, namely determine an instance of a mathematical object per class and measure closeness of the layerwise output of an incoming datapoint to these objects with the prediction corresponding to the class whose object was closest in this sense. In contrast to (Alain & Bengio, 2018), where parameters of their linear probe are learned iteratively by minimizing a cross-entropy loss, we seek to determine the linear probe parameters directly via the geometry of the class-conditional training data. We choose to simply use a vector as this mathematical object and measure closeness in the angle sense. We call this probe, the Vector Linear Probe Intermediate-layer Classifier (VeLPIC). As we discuss subsequently, we find, surprisingly, that this choice is significantly more effective than MASC, in most cases. Secondly, we show that we can use the parameters of the probe as applied to the last layer, to modify the model weights to immediately confer the corresponding generalization to the model.

We now discuss how the vector corresponding to each class in VeLPIC is constructed. Each class vector is determined using only the top principal component from PCA run on augmented[2] class-conditional corrupted training data. However, the first principal component can manifest in two opposite directions (i.e. the vector or its negative). This is important here[3] because incoming data vectors can be "close" to this class vector, even though their angles are obtuse and closer to $180°$. VeLPIC resolves this directional issue by aligning the class vector based on the sign of the projection of the training data mean; if the mean of the training data projected on this principal component is negative, the direction of the principal component is flipped to obtain the class vector; otherwise, it is retained as is.

---

**Algorithm 1 Vector Linear Probe Intermediate-layer Classifier (VeLPIC)**

**Input:** Principal component vectors $\{\mathcal{P}_m\}_{m=1}^M$, projection training means $\{T_m\}_{m=1}^M$, layer $l$ output $\boldsymbol{x_l}$, class labels $\{C_m\}_{m=1}^M$
**Output:** Predicted label $y(\boldsymbol{x_l})$
1: **for** each class $m = 1, \ldots, M$ **do**
2:     **if** $T_m < 0$ **then**
3:        $\mathcal{V}_m \leftarrow -\mathcal{P}_m$
4:     **else**
5:        $\mathcal{V}_m \leftarrow \mathcal{P}_m$
6:     **end if**
7: **end for**
8: **for** each class $m = 1, \ldots, M$ **do**
9:     $\boldsymbol{x_{lm}} \leftarrow$ Projection of $\boldsymbol{x_l}$ onto $\mathcal{V}_m$
10: **end for**
11: $y(\boldsymbol{x_l}) \leftarrow C_j$ where $j = \arg\max_m \boldsymbol{x_{lm}}$
12: **Return:** $y(\boldsymbol{x_l})$

---

Formally, for a given test data point $\boldsymbol{x}$, let $\boldsymbol{x_l}$ denote its activation at layer $l$ obtained in the forward pass of $\boldsymbol{x}$ through the Deep Network until the output of layer $l$. For layer $l$, let $\{\mathcal{P}_m\}_{m=1}^M$ be the top principal component vectors, one each per class, of the class-conditional corrupted training data and $\{T_m\}_{m=1}^M$ be its corresponding[4] projection means, where $M$ is the number of classes. Let $\{\mathcal{V}_m\}_{m=1}^M$ be unit vectors representing VeLPIC class vectors. VeLPIC uses $\{\mathcal{V}_m\}_{m=1}^M$ to predict the label of $\boldsymbol{x_l}$ based on its maximum projection among these class vectors[5], as outlined in Algorithm 1.

---

[2]We augment class training data points with their negative, so as to obtain a 1-D subspace, rather than a 1-D affine space, along the lines of the subspace construction procedure for MASC.

[3]Observe that this isn't an issue with MASC, since it is a quadratic classifier.

[4]i.e. $T_i$ is the mean of projecting training data points on $\mathcal{P}_i$.

[5]This is equivalent to minimum angle to the VeLPIC class vectors.

## 5.1 TRAINING DYNAMICS OF THE LINEAR PROBE

Here, we examine if a linear probe (i.e. VeLPIC) can decode latent generalization with performance comparable to MASC. To this end, we tracked the performance of VeLPIC, during training.

VeLPIC test accuracy during training for MLP-MNIST, CNN-Fashion-MNIST and AlexNet-Tiny ImageNet are shown in Figure 2. Results with 0% and 100% corruption degrees as well as the results with additional models are shown in Figure 7 and 8, respectively in Appendix Section C.

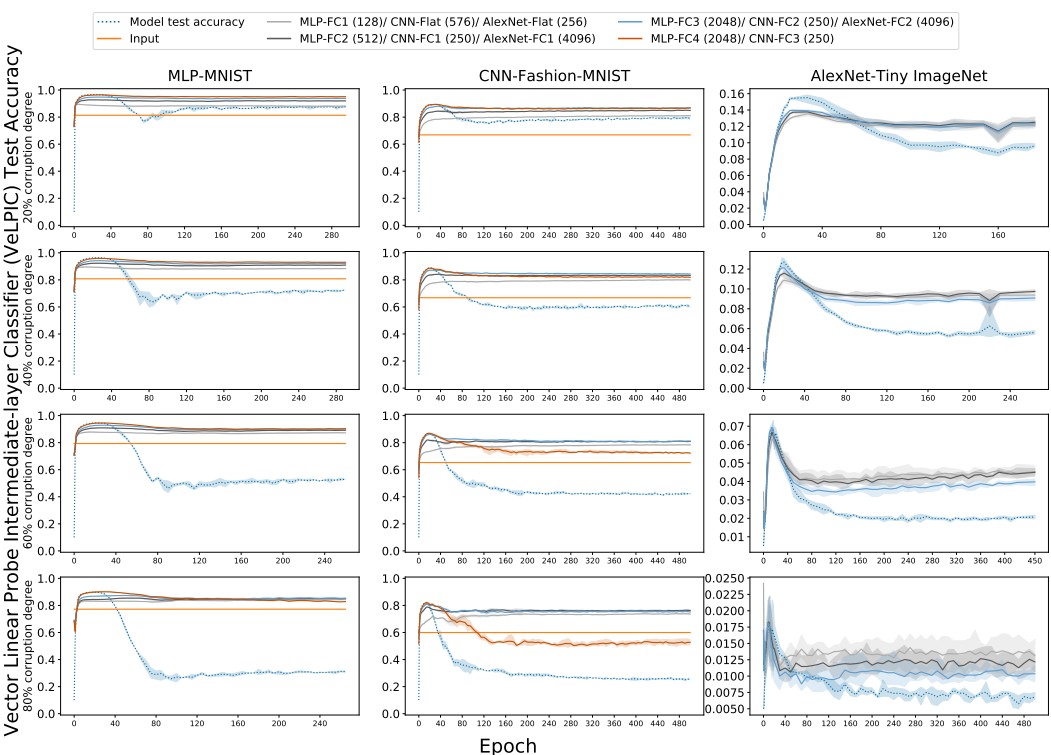

Figure 2: Vector Linear Probe Intermediate-layer Classifier (VeLPIC) test accuracy during training of the network, where test data is projected onto class vectors constructed at each epoch from training data with the indicated label corruption degrees. The plots display VeLPIC accuracy across different layers of the network for various model–dataset combinations. For reference, the test accuracy of the models (blue dotted line) over epochs of training is also shown. FC denotes fully connected layers with $ReLU$ activation, and Flat refers to the flatten layer without $ReLU$.

Unexpectedly, not only does VeLPIC not perform worse than MASC, we find indeed that VeLPIC almost always significantly outperforms MASC, especially with higher corruption degrees. In fact, for representations from many layers, VeLPIC is able to extract significantly better latent generalization performance than MASC and our results show that, for these layers, VeLPIC's performance plateaus at significantly higher levels than MASC. The difference between VeLPIC test accuracy and MASC test accuracy are shown in Figure 9, 10 & 11 in the Appendix Section C.1.

## 6 TRANSFERRING LATENT GENERALIZATION TO MODEL GENERALIZATION

Here, we ask if the latent generalization in models that memorize, can be directly transferred to the model, in order to immediately improve its generalization. To this end, it turns out that the class vectors of VeLPIC applied to the last layer can be directly substituted in the pre-softmax layer of the model as an intervention that transfers VeLPIC's generalization performance to the model, without further training. We elaborate below on how this is so.

Consider a model whose last layer (i.e. the layer preceding the pre-softmax layer) consists of $d$ units. Let $\boldsymbol{v}_j \in \mathbb{R}^d$ be the VeLPIC class vector for class $j$. The new pre-softmax weight matrix $\mathbf{W}_{\text{pre-softmax}} \in \mathbb{R}^{M \times d}$ is constructed as:

$$\mathbf{W}_{\text{pre-softmax}} = \left( \begin{bmatrix} \boldsymbol{v}_1 & \boldsymbol{v}_2 & \cdots & \boldsymbol{v}_M \end{bmatrix} \right)^{\top} \tag{3}$$

This weight matrix $\mathbf{W}_{\text{pre-softmax}}$ replaces the original pre-softmax weights, and all biases are set to zero. It is straightforward to see that this substitution results in the model making the same predictions as VeLPIC applied to the last layer.

During model training, we replace the pre-softmax weights with VeLPIC vectors, as indicated above and evaluate the model's performance on the test dataset at each epoch. Figure 3 presents these results for MLP-MNIST, CNN-Fashion-MNIST, and AlexNet-Tiny ImageNet. Additional results, including models with 0% and 100% corruption levels, and other model-dataset pairs are presented in Figure 12 and Figure 13, respectively in the Appendix Section D.

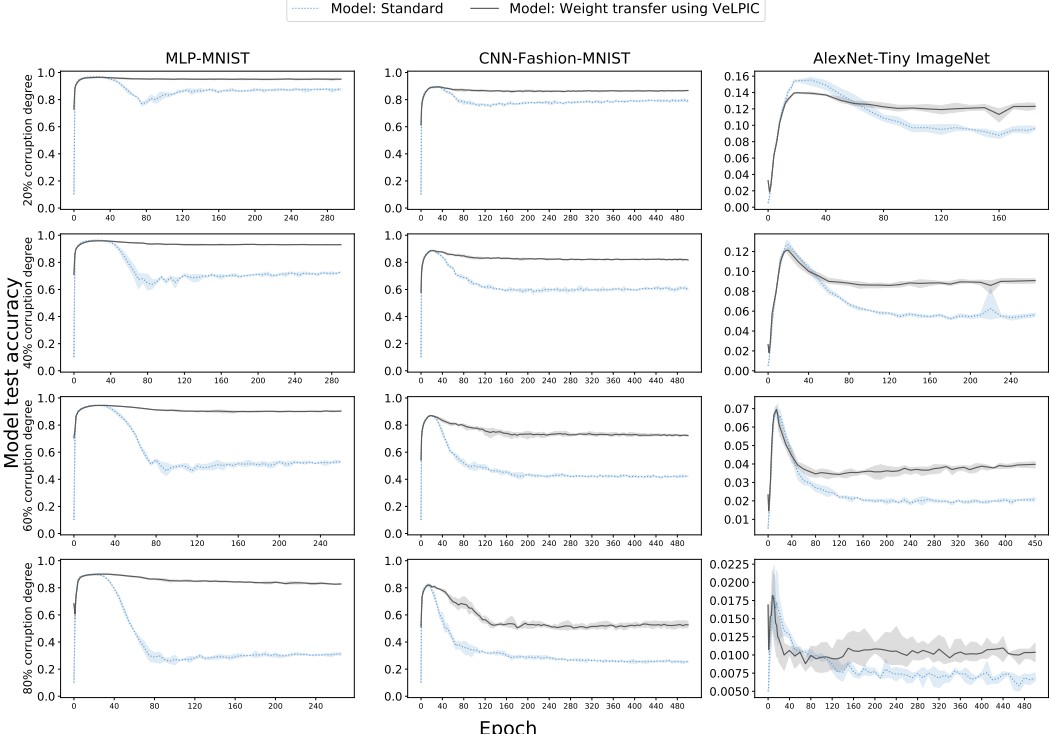

Figure 3: Model test accuracy when the weight intervention is applied to the epoch in question during training. The test accuracy of the model with standard training without weight intervention (blue dotted line) is overlaid for comparison.

We observe that the weight intervention that replaces pre-softmax weights with the VeLPIC vectors leads to an immediate & significant improvement in generalization performance in every epoch of the latter phase of training, matching that of the linear probe, & notably without any further training. This establishes that the latent generalization in memorized models can be directly harnessed to enhance their test performance, even in the presence of label noise.

## 7 MEMORIZATION-RESISTANT INITIALIZATIONS

It is thought (Stephenson et al., 2021) that avoiding[6] large-scale memorization of training data could play a key role in causing Deep Networks to generalize well. An important question, therefore, is

---

[6]It has also been suggested(Feldman & Zhang, 2020) that some memorization could help with generalization, when the data distribution is long-tailed.

whether we can make Deep Networks avoid memorization even in the corrupted labels setting we study here, and if doing so can improve their generalization performance.

Here, leveraging the linear probe that we built, we explored a new initialization strategy for Deep Networks that tends to avoid memorizing training data. Specifically, we start with a random initialization of all weights, and construct last layer VeLPIC class vectors for the randomly-initialized network. We then substitute these class vectors onto the pre-softmax weights, as outlined in the previous section. The rest of the model remains randomly initialized. Standard training with gradient clipping & reduced[7] learning rate by factor of 10 is performed using the corrupted training dataset for 100 epochs.

For MLP-MNIST, MLP-CIFAR10, CNN-MNIST and CNN-Fashion-MNIST, we track & report the model's training accuracy on corrupted labels and test accuracy on true labels across training epochs for varying corruption degrees, in Figure 4, where we also overlay (dotted lines) the dynamics of the test accuracy of the model with standard training[8] on standard initialization. In Appendix Section E.2, we also present training dynamics of the model over epochs of training, separately for the subset of training data whose labels were flipped during the corruption process, and for the subset of training data points whose labels remain uncorrupted.

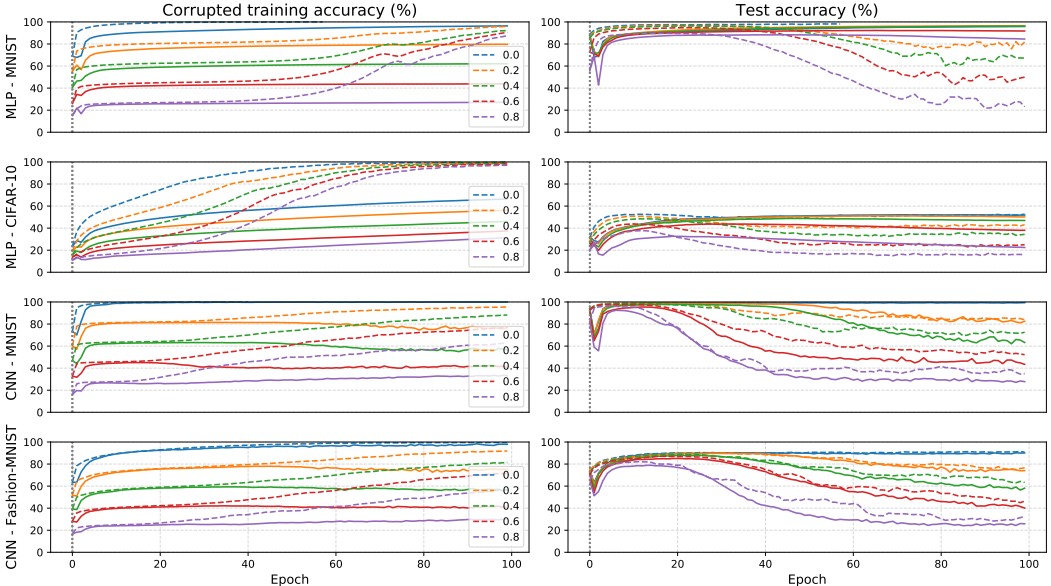

Figure 4: Model train accuracy with corrupted labels and model test accuracy with true labels during training when intervention is performed at random initialization and standard training is performed thereafter with gradient clipping. The standard training model (dotted), trained without gradient clipping, with a $10\times$ higher learning rate, and without intervention, is overlaid for comparison.

We find that, for most models, this initialization strategy is effective in causing the model to resist memorization, i.e. the model refrains from correctly learning a large fraction of corrupted training labels. For MLPs, this act of resisting memorization is also accompanied by significantly better generalization over epochs of training, in comparison to standard training with standard initializations. For CNNs, this resistance to memorization is accompanied by generalization performance comparable to standard training with standard initialization. Strikingly, for many of these models, there is little degradation in generalization performance near the initial epochs, unlike what we observe with generalization in models with standard initializations.

---

[7]than ones mentioned in the Appendix Section A

[8]The models were trained without gradient clipping, using the learning rate specified in the Appendix Section A.

To our knowledge, this is the first report of an initialization strategy being effective in resisting memorization. As such, we believe that understanding the mechanisms underlying the effectiveness of these initializations represents an important and promising direction for future investigation.

We also conduct experiments using standard training without applying gradient clipping or reducing the learning rate. These results are sometimes brittle, in that their generalization performance in some cases suddenly drops to chance level. Furthermore, in some cases, these initializations are not effective in resisting memorization (Appendix Section E.3). In the case of standard training only, we also perform intervention experiments by modifying the pre-softmax weights at the 10th and 40th epochs; the corresponding results are reported in the Appendix. Additionally, the Appendix presents a comparison of best-layer MASC accuracy, best-layer VeLPIC accuracy, and models trained with memorization-resistant initialization across different corruption levels.

## 8 DISCUSSION

The notion of memorization, where Deep Networks are able to perfectly learn noisy data at the expense of generalization has posed a challenge to traditional notions of generalization from Statistical Learning Theory (Zhang et al., 2017; 2021). Recent work (Ketha & Ramaswamy, 2025) demonstrating improved latent generalization in such models is an interesting new development in our understanding of memorization and the nature of representations that drive it. Our goal here was to take a deeper dive into this phenomenon, to investigate the origin and dynamics of latent generalization. While the dynamics of memorization and generalization early in training have seen detailed empirical investigation (Arpit et al., 2017), the phenomenon of fall in model generalization in the later phase of training is more poorly understood. We showed that early-on in training, latent generalization and the model's generalization closely follow each other, suggesting common mechanisms that contribute to both. However, later in training, there is a divergence, with the model retaining significant latent generalization ability, while sacrificing overt model generalization to a greater degree. After showing that MASC (Ketha & Ramaswamy, 2025) is a quadratic classifier, we built a new linear probe (VeLPIC) and found, unexpectedly, that it has better latent generalization performance in comparison to MASC, in most cases. Indeed, while (Ketha & Ramaswamy, 2025) show that MASC applied to at least one layer outperforms the model at the end of training, with respect to generalization, with VeLPIC, we find that, in most cases, all layers' latent generalization outperform model generalization. This implies that the latent generalization effect during memorization is more pronounced and more widely present among layer representations than previously reported in (Ketha & Ramaswamy, 2025). We were also interested in examining if the latent generalization could readily be translated to model generalization by directly modifying model weights. We utilized the linear probe to derive a new set of model pre-softmax weights to make this so. Finally, we leveraged this understanding to create new kinds of initializations for Deep Networks and show that they resist memorization in favor of generalization, in many cases. These results point to the possibility of the existence of a different part of the loss landscape that is more effective in avoiding memorization, and as such, merits more detailed investigation.

This work brings up multiple new directions for investigation. While we have made some progress, the detailed mechanisms governing latent generalization during memorization remain to be investigated. It is also an open question, whether there exist other probes that can extract better latent generalization from layerwise representations, in comparison to MASC and VeLPIC. Next, it is unclear if latent generalization from representations of layers other than the last layer can be transferred towards model generalization. This can be useful to do, in cases where early or middle layers exhibit better latent generalization than the last layer. Also, it is worth examining, if the memorization-resistant initializations proposed here can be further refined. It remains to be examined why these initializations are extraordinarily effective in fending off memorization, especially in the latter epochs of training. More generally, in light of these results, whether an understanding of generalization in the memorization regime can inform a better understanding of generalization for models trained with uncorrupted labels is a worthwhile direction for future investigation.

In closing, our results highlight the rich role of representations in driving generalization during memorization, how their understanding can be utilized to directly improve model generalization and in order to design memorization-resistant initializations for Deep Networks, in the memorization regime.

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

## A  EXPERIMENTAL SETUP

We demonstrate results for the same set of models and datasets as presented in (Ketha & Ramaswamy, 2025). Specifically, we use Multi-Layer Perceptrons (MLPs) trained on the MNIST (Deng, 2012) and CIFAR-10 (Krizhevsky, 2009) datasets; Convolutional Neural Networks (CNNs) trained on MNIST, Fashion-MNIST (Xiao et al., 2017), and CIFAR-10; and AlexNet (Krizhevsky et al., 2012) trained on Tiny ImageNet dataset (Moustafa, 2017).

Each model was trained under two distinct schemes: (i) using training data with true labels, referred to as "generalized models," and (ii) using training data with labels randomly shuffled to varying degrees (referred to as "memorized models" Zhang et al. (2021). Similar to Ketha & Ramaswamy (2025), we train the aforementioned models using corruption degrees of 0%, 20% , 40%, 60%, 80%, and 100%. Training with a corruption degree $c$ implies that, with probability $c$, the label of a training datapoint is changed with a randomly selected label drawn uniformly from the set of possible classes. This may result in the label remaining same after the change as well. All models were trained either until achieving high training accuracy (99% or 100%) or for a maximum of 500 epochs, whichever occurred first.

To study the dynamics of the training process, we conducted the experiments on model checkpoints saved at various stages of training. Specifically, we began with the randomly initialized model (corresponding to epoch 0), followed by checkpoints saved at every second epoch up to the 20th epoch. Beyond epoch 20, results are shown at intervals of five epochs for the MLP and CNN models, and at intervals of ten epochs for the AlexNet model. The reported results are averaged over three independent training runs, with shaded regions in the plots indicating the range across instances. We have used 99% as the percentage of variance explained by the principal components that form the class-specific subspaces used by MASC, similar to experiments conducted in Ketha & Ramaswamy (2025).

The experiments were conducted on servers and workstations equipped with NVIDIA GeForce RTX 3080, RTX 3090, Tesla V100, and Tesla A100 GPUs. The server runs on Rocky Linux 8.10 (Green Obsidian), while the workstation uses Ubuntu 20.04.3 LTS. Memory requirements varied depending on the specific experiments and models. All model implementations were developed in Python using the PyTorch library, with torch.manual_seed set to 42 to ensure reproducibility. Accuracy served as the primary evaluation metric throughout this work.

### A.1  MODEL ARCHITECTURES AND TRAINING DETAILS

**MLP Model.** The MLP architecture consists of four hidden layers with 128, 512, 2048, and 2048 units, respectively. Each layer is followed by a $ReLU$ activation, and a $softmax$ layer is used for classification. Models were trained SGD Qian (1999) with a learning rate of $1 \times 10^{-3}$ and momentum 0.9. A batch size of 32 was used across all experiments. Input dataset was normalized by dividing pixel values by 255.

**CNN Model.** The CNN model[9] is composed of three convolutional blocks, each containing two convolutional layers followed by a max pooling layer. The convolutional layers use 16, 32, and 64 filters, respectively, with kernel size $3 \times 3$ and stride 1. The max pooling layers have a kernel size of $2 \times 2$ and stride 1. These blocks are followed by three fully connected layers with 250 units each. ReLU activation is used after all layers except pooling, and softmax is used at the output for classification. The CNN was trained using Adam optimizer Kingma (2014) with a learning rate of 0.0002. For MNIST and Fashion-MNIST, a batch size of 32 was used, while for CIFAR-10, a batch size of 128 was used. Input data was normalized by subtracting the mean and dividing by the standard deviation of each channel.

### A.2  MINIMUM ANGLE SUBSPACE CLASSIFIER

We summarize below the Minimum Angle Subspace Classifier (MASC) from (Ketha & Ramaswamy, 2025), in order to keep the exposition here largely self-contained.

---

[9]The convolution network were implemented following the design principals outlined in (Tran et al., 2022).

For a given Deep Network, MASC leverages the class-specific geometric structure of network's latent representations. For an input data point $x$, let its activation vector at layer $l$ be denoted by $x_l$. The objective is to classify $x_l$ by leveraging a set of class-conditional subspaces, $\{S_k\}_{k=1}^K$, estimated from a training dataset $\mathcal{D} = \{(x_i, y_i)\}_{i=1}^m$. To predict the class label $y(x_l)$, MASC Algorithm 2 (reproduced verbatim from (Ketha & Ramaswamy, 2025)), assigns $x_l$ to the class whose training subspace forms the smallest angle with it.

The class-conditional subspaces $\{S_k\}_{k=1}^K$ are estimated from the training dataset $\mathcal{D} = \{(x_i, y_i)\}_{i=1}^m$, where each $x_i \in \mathbb{R}^d$ is paired with a label $y_i \in \{C_k\}_{k=1}^K$. For a given layer $l$, these subspaces are constructed following Algorithms 3 and 4 (reproduced verbatim from (Ketha & Ramaswamy, 2025)). In practice, each subspace $S_k$ is represented by its principal components, which provide a compact basis for capturing the underlying class-conditional structure.

---

**Algorithm 2 Minimum Angle Subspace Classifier (MASC)** (reproduced verbatim from (Ketha & Ramaswamy, 2025))

---

1: **Input:** Training subspaces $\{S_k\}_{k=1}^K$, layer output data point $x_l$ from layer $l$ when input $x$ is passed through the network and classes $\{C_k\}_{k=1}^K$.
2: **Output:** MASC prediction class label $y(x_l)$ according to layer $l$ .
3: **for** each class $C_k$ **do**
4:    $x_{lk} \longleftarrow$ compute the projection of $x_l$ onto subspace $S_k$.
5:    Compute the angle $\theta(x_l, x_{lk})$ between $x_l$ and $x_{lk}$
6: **end for**
7: Assign the label $y(x_l) = C_k$ where $k = \arg\min_k \theta(x_l, x_{lk})$

---

**Algorithm 3 Subspaces Estimator for MASC**
(reproduced verbatim from (Ketha & Ramaswamy, 2025))

---

1: **Input:** Training dataset $\mathcal{D}\{(x_i, y_i)\}_{i=1}^m \in \mathbb{R}^d \times \mathbb{R}$, where each $x_i \in \mathbb{R}^d$ and $y_i \in \{C_k\}_{k=1}^K$ are input-label pairs, neural network, and layer $l$.
2: **Output:** Subspaces $\{S_k\}_{k=1}^K$ for classes $K$ and given layer $l$.
3: $\mathcal{D}_l = \phi$
4: **for** each input pair $(x_i, y_i)$ in $\mathcal{D}$ **do**
5:    Pass $x_i$ through the network layers to obtain the output of layer $l$, denoted as $x_l \in \mathbb{R}^{ld}$.
6:    $\mathcal{D}_l = \mathcal{D}_l \cup \{x_l\}$
7: **end for**
8: Estimated subspaces $\{S_k\}_{k=1}^K \longleftarrow$ **PCA-Based Subspace Estimation**$(\mathcal{D}_l)$
9: **Return:** Subspaces $\{S_k\}_{k=1}^K$

---

**Algorithm 4 PCA-Based Subspace Estimation**
(reproduced verbatim from (Ketha & Ramaswamy, 2025))

---

1: **Input:** Layer output $\mathcal{D}_l = \{(x_i, y_i)\}_{i=1}^m$, where $x_l \in \mathbb{R}^{ld}$ and $y_i \in \{C_k\}_{k=1}^K$.
2: **Output:** Subspaces $\{S_k\}_{k=1}^K$ for classes $K$.
3: $\mathcal{D}_{\text{new}} \leftarrow \mathcal{D}_l$
4: **for** each data point $x_l$ in $\mathcal{D}_l$ **do**
5:    $\mathcal{D}_{\text{new}} \leftarrow \mathcal{D}_{\text{new}} \cup \{-x_l\}$
6: **end for**
7: **for** each class $C_k$ in $C_K$ **do**
8:    Extract the subset of data $\mathcal{D}_{\text{new},k} = \{x_l \mid y_i = k\}$
9:    Apply PCA to $\mathcal{D}_{\text{new},k}$ to calculate the PCA components
10:    The span of the PCA components defines the subspace $S_k$
11: **end for**
12: **Return:** Subspaces $\{S_k\}_{k=1}^K$

---

## B  TRAINING DYNAMICS OF LATENT GENERALIZATION USING MASC

MASC testing accuracy during training for MLP trained on MNIST, CNN trained on Fashion-MNIST and AlexNet trained on Tiny ImageNet with 0% and 100% corruption degrees are shown in Figure 5. In the absence of label corruption, the MASC accuracy of the final fully connected layers (MLP-FC4 (2048 units) and CNN-FC3 (250 units)) closely matched the corresponding model test accuracies. Interestingly, in certain cases, such as AlexNet trained on Tiny ImageNet (FC1 and FC2, each with 4096 units) and MLP-CIFAR-10 (FC3 with 2048 units), the MASC accuracy even surpassed the model's test accuracy. Results with additional models i.e. MLP trained on CIFAR-10, CNN trained on MNIST, CNN trained on CIFAR-10 for various corruption degrees are shown in Figure 6.

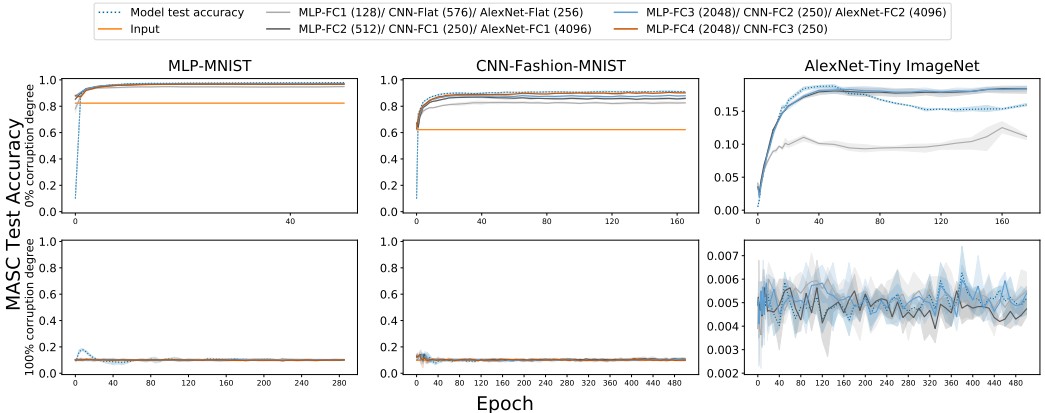

Figure 5: Minimum Angle Subspace Classifier (MASC) Test accuracy for 0% and 100% corruption degrees during training of the network, where test data is projected onto class-specific subspaces constructed from training data with the indicated label corruption degrees. The plots display MASC accuracy across different layers of the network for various model–dataset combinations. For reference, the test accuracy of the models (dotted line) is also shown. Each row corresponds to a specific corruption degree, while columns represent different models, as labeled. FC denotes fully connected layers with $ReLU$ activation, and Flat refers to the flatten layer without $ReLU$.

## C  TRAINING DYNAMICS OF THE LINEAR PROBE: VELPIC

A linear probe – VeLPIC – test accuracy during training for MLP-MNIST, CNN-Fashion-MNIST and AlexNet-Tiny ImageNet with 0% and 100% corruption degrees are shown in Figure 7. Results with additional models i.e. MLP-CIFAR-10, CNN-MNIST, CNN-CIFAR-10 for various corruption degrees are shown in Figure 8.

### C.1  DIFFERENCE BETWEEN VELPIC AND MASC

Here, we present the difference between test accuracy of VeLPIC and MASC during training and for different layer of the networks. For MLP-MNIST, CNN-Fashion-MNIST and AlexNet-Tiny ImageNet, these results are shown in Figure 9 and Figure 10. Results with additional models i.e. MLP-CIFAR-10, CNN-MNIST, CNN-CIFAR-10 for various corruption degrees are shown in Figure 11.

## D  TRANSFERRING LATENT GENERALIZATION TO MODEL GENERALIZATION

For MLP-MNIST, CNN-Fashion-MNIST and AlexNet-Tiny ImageNet with 0% and 100% corruption degrees, model test accuracy during training when we replace the pre-softmax weights with VeLPIC vectors are shown in Figure 12. Results with additional models i.e. MLP-CIFAR-10, CNN-

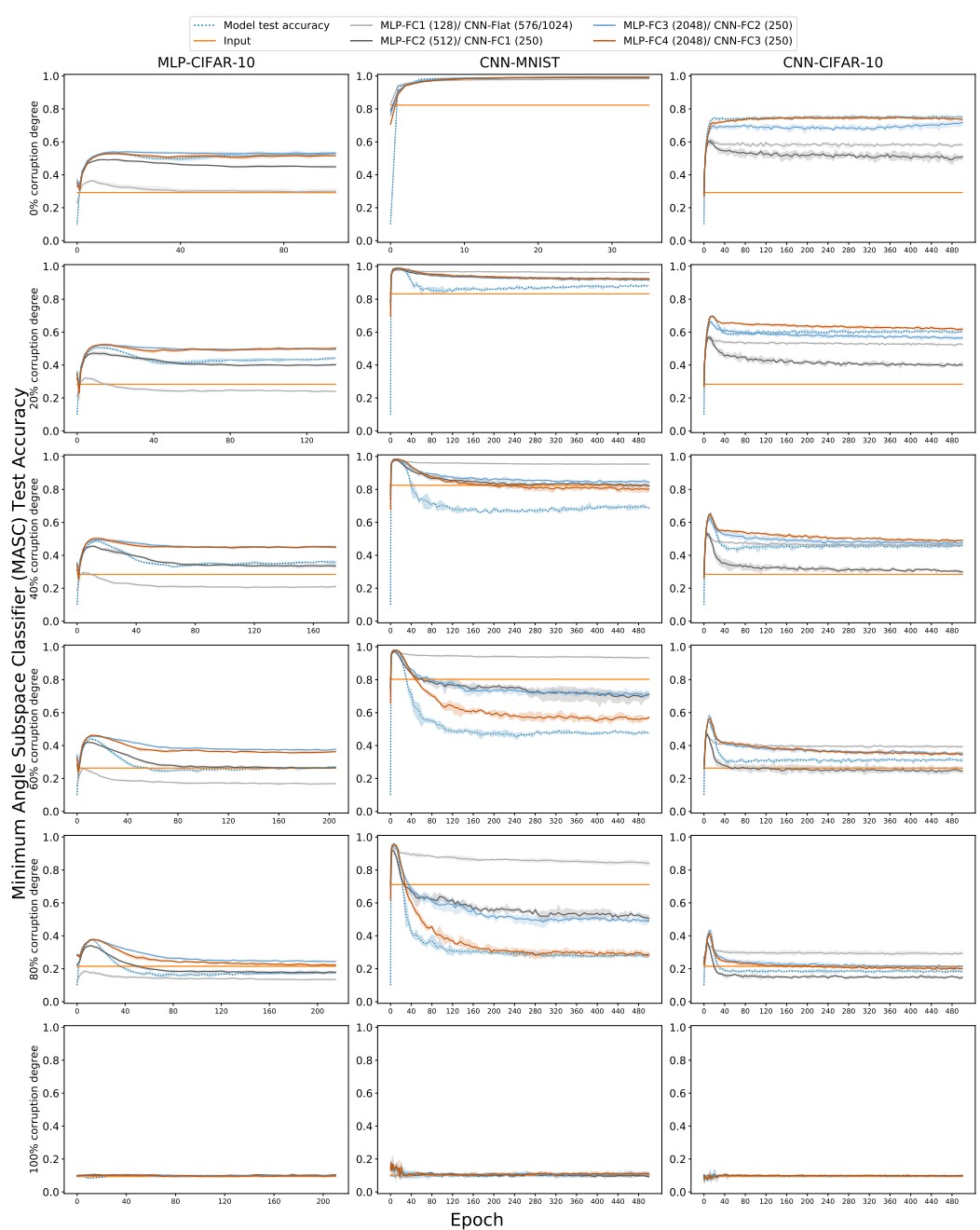

Figure 6: MASC accuracy during training of the network, where test data is projected onto class-specific subspaces constructed from training data with the indicated label corruption degrees. The plots display MASC accuracy across different layers of the network for various model–dataset combinations. For reference, the test accuracy of the models (dotted line) is also shown. Each row corresponds to a specific corruption degree, while columns represent different models, as labeled. FC denotes fully connected layers with $ReLU$ activation, and Flat refers to the flatten layer without $ReLU$.

MNIST, CNN-CIFAR-10 for various corruption degrees are shown in Figure 13. Model corrupted training accuracy for all models-dataset-corruption are plotted in Figure 14 and Figure 15.

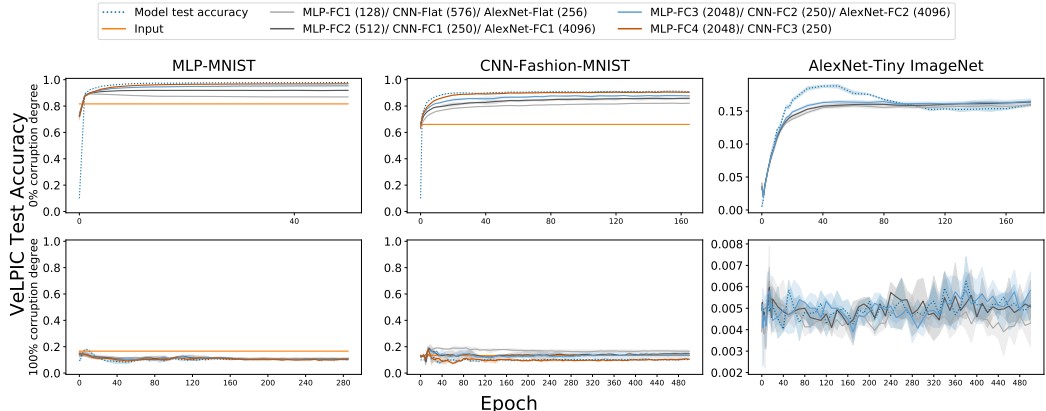

Figure 7: Vector Linear Probe Intermediate-layer Classifier (VeLPIC) test accuracy for 0% and 100% corruption degrees during training of the network, where test data is projected onto class vectors constructed at each epoch from training data with the indicated label corruption degrees. The plots display VeLPIC accuracy across different layers of the network for various model–dataset combinations. For reference, the test accuracy of the models (blue dotted line) over epochs of training is also shown. FC denotes fully connected layers with $ReLU$ activation, and Flat refers to the flatten layer without $ReLU$.

# E   MEMORIZATION-RESISTANT INITIALIZATIONS

## E.1   STANDARD RESULTS

In Figure 16, we conducted standard training on the corrupted datasets without any interventions and see results consistent with those reported in Arpit et al. (2017).

## E.2   RESULTS OF FLIPPED AND UN-FLIPPED ACCURACIES

We here present training dynamics of the model over epochs of training, separately for the subset of training data whose labels were flipped during the corruption process (flipped accuracy), and for the subset of training data points whose labels remain uncorrupted (unflipped accuracy). We present results for models where the intervention is applied by replacing the pre-softmax layer weights with VeLPIC class vectors at random initialization (see Figure 17).

For memorization-resistant initialization (Figure 17), we find that the training accuracy on flipped labels remains low, while the accuracy on unflipped (i.e. true labels) labels is typically high. This suggests that this initialization has a tendency to resist memorization.

## E.3   MEMORIZATION-RESISTANT INITIALIZATIONS: WITH STANDARD TRAINING ONLY

Here, we show the results of memorization-resistant initializations with standard training only. Specifically, we begin with random weight initialization and construct last-layer VeLPIC class vectors for the randomly initialized network. These vectors are then substituted into the pre-softmax weights, while the rest of the model remains randomly initialized. The network is then trained on the corrupted dataset for 100 epochs using standard training.

Model's training accuracy on corrupted labels and test accuracy on true labels when intervention is performed at random initialization across training epochs for MLP-MNIST, CNN-MNIST and CNN-Fashion-MNIST are shown in Figure 18.

For additional models and varying corruption degrees, model's training accuracy on corrupted labels and test accuracy on true labels when intervention is performed at random initialization across training epochs are shown in Figure 19.

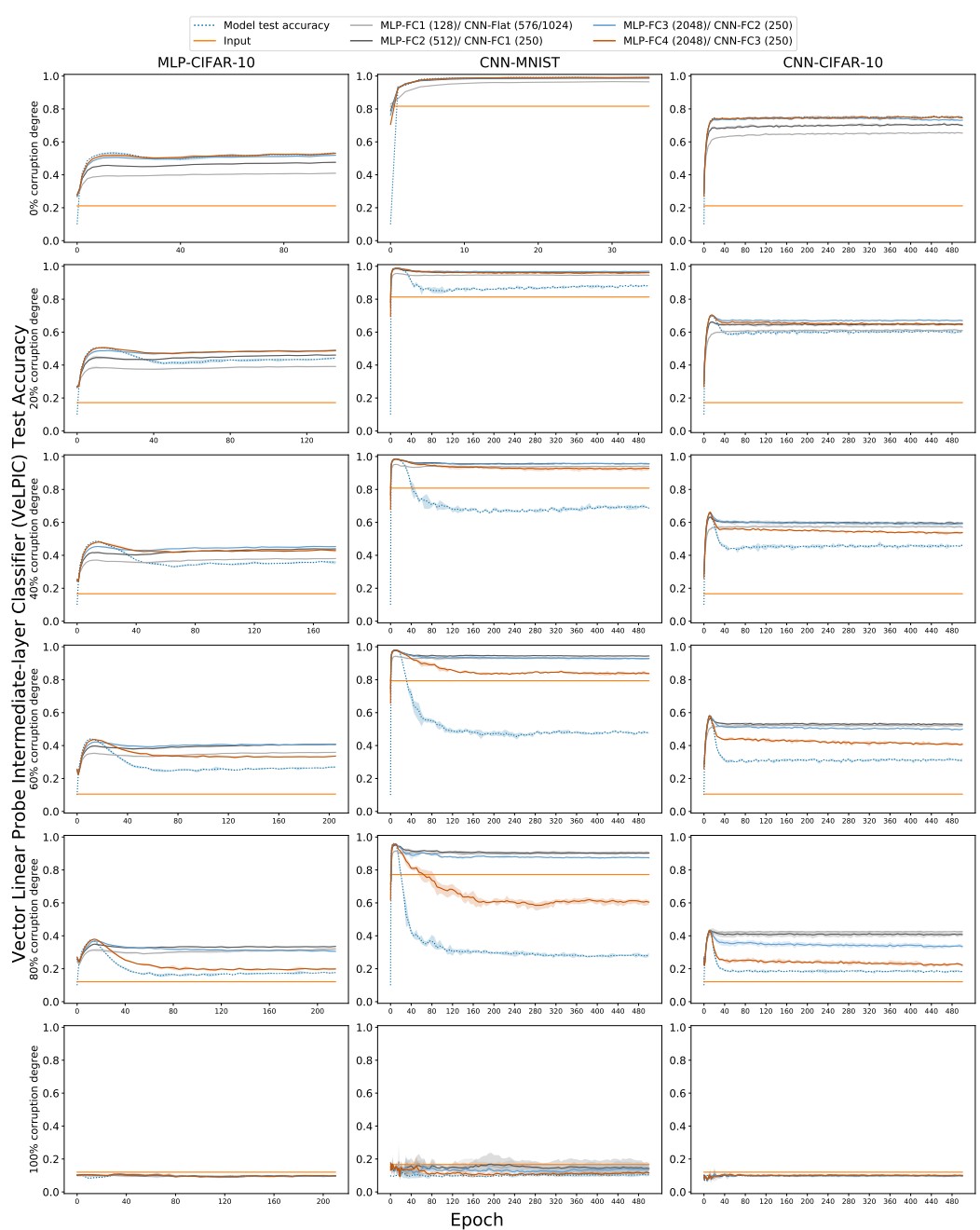

Figure 8: Vector Linear Probe Intermediate-layer Classifier (VeLPIC) test accuracy during training of the network, where test data is projected onto class vectors constructed at each epoch from training data with the indicated label corruption degrees. The plots display VeLPIC accuracy across different layers of the network for various model–dataset combinations. For reference, the test accuracy of the models (blue dotted line) over epochs of training is also shown. FC denotes fully connected layers with $ReLU$ activation, and Flat refers to the flatten layer without $ReLU$.

### E.3.1 COMPARISON RESULTS

Comparison of model's test accuracy for various corruption degrees are shown in Figure 20. The model at the 100th epoch was used for comparison; if unavailable due to earlier training termination, the final trained model was used instead. For comparison, we use the best-layer MASC test accuracy

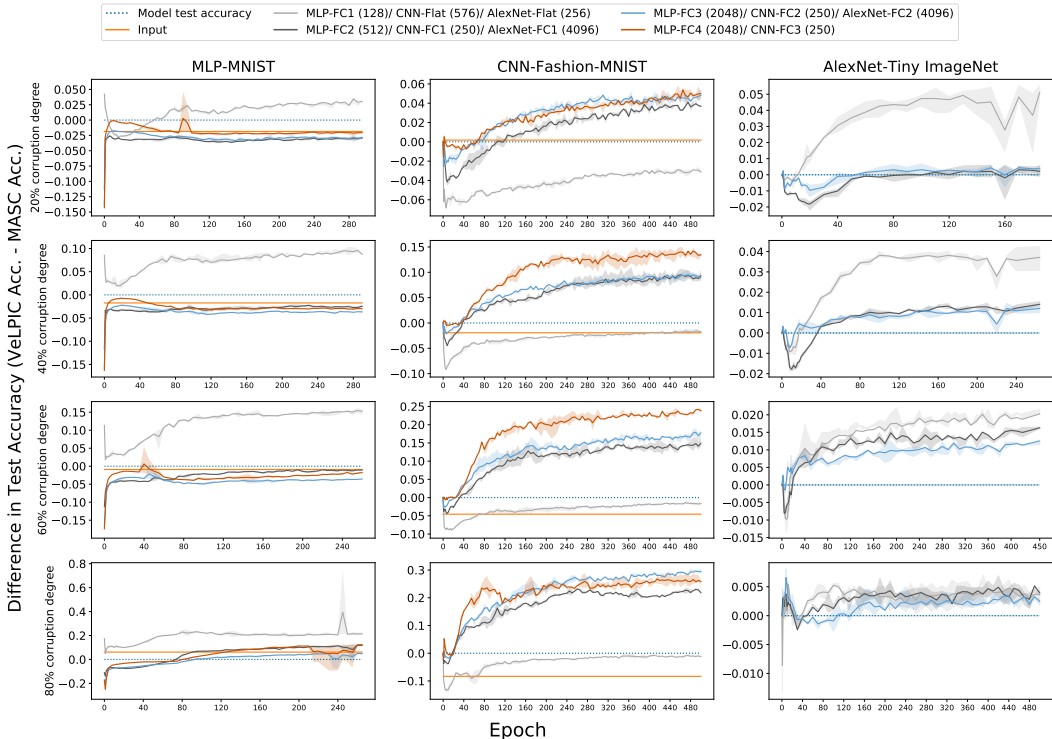

Figure 9: Difference in test accuracy (VeLPIC Accuracy - MASC Accuracy) during training of the network, where test data is projected onto class vectors constructed at each epoch from training data with the indicated label corruption degrees. The plots display difference in accuracy across different layers of the network for various model–dataset combinations. For reference, the test accuracy of the models (blue dotted line) over epochs of training is also shown, which would be 0.

(with subspaces capturing 99% variance), the best-layer VeLPIC test accuracy, and the test accuracy after applying the intervention at random initialization. For models with the intervention, two values are reported: the test accuracy at the 100th epoch and the maximum test accuracy achieved during training.

### E.3.2 EXPERIMENT RESULTS WITH WEIGHT INTERVENTION AT 10TH EPOCH

Here, we present results of weight intervention applied at 10th epoch. The model is training with corrupted data for the first 10 epochs using standard training. The intervention is performed at 10th epoch by replacing the pre-softmax weights with VeLPIC vectors (last layer). Standard training is performed for the next 90 epochs using corrupted training data. Model's training accuracy on corrupted labels and test accuracy on true labels when intervention is performed at 10th epoch across training epochs for varying corruption degrees are shown in Figure 21.

Briefly, we find that the intervention seems to switch the training dynamics to a regime where it resists memorization, following the intervention, which is accompanied by better model generalization. Also, the generalization dynamics in these cases appears to be less brittle and is effective in some models where the initializations did not demonstrate effectiveness.

### E.3.3 EXPERIMENT RESULTS WITH WEIGHT INTERVENTION AT 40TH EPOCH

Similarly to the previous section, here we present results of the weight intervention. The intervention is performed at 40th epoch. The model is training with corrupted data for the first 40 epochs using standard training. The intervention is performed at 40th epoch by replacing the pre-softmax weights with VeLPIC vectors (last layer). Standard training is performed for the next 60 epochs us-

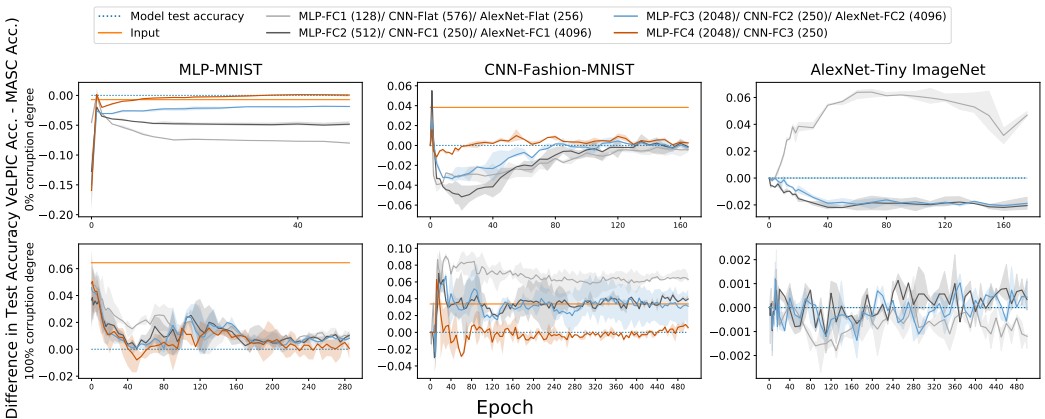

Figure 10: Difference in test accuracy (VeLPIC Accuracy - MASC Accuracy) during training of the network, where test data is projected onto class vectors constructed at each epoch from training data with the indicated label corruption degrees. The plots display difference in accuracy across different layers of the network for various model–dataset combinations. For reference, the test accuracy of the models (blue dotted line) over epochs of training is also shown, which would be 0.

ing corrupted training data. Model's training accuracy on corrupted labels and test accuracy on true labels when intervention is performed at 40th epoch across training epochs for varying corruption degrees are shown in Figure 22.

### E.3.4 RESULTS OF FLIPPED AND UN-FLIPPED ACCURACIES

We here present training dynamics of the model over epochs of training, separately for the subset of training data whose labels were flipped during the corruption process (flipped accuracy), and for the subset of training data points whose labels remain uncorrupted (unflipped accuracy). We present the above results for different models when interventions is applied at three points where the pre-softmax layer weights are replaced with the VeLPIC class vectors: at random initialization (see Figure 23), at the 10th epoch (Figure 24), and at the 40th epoch (Figure 25).

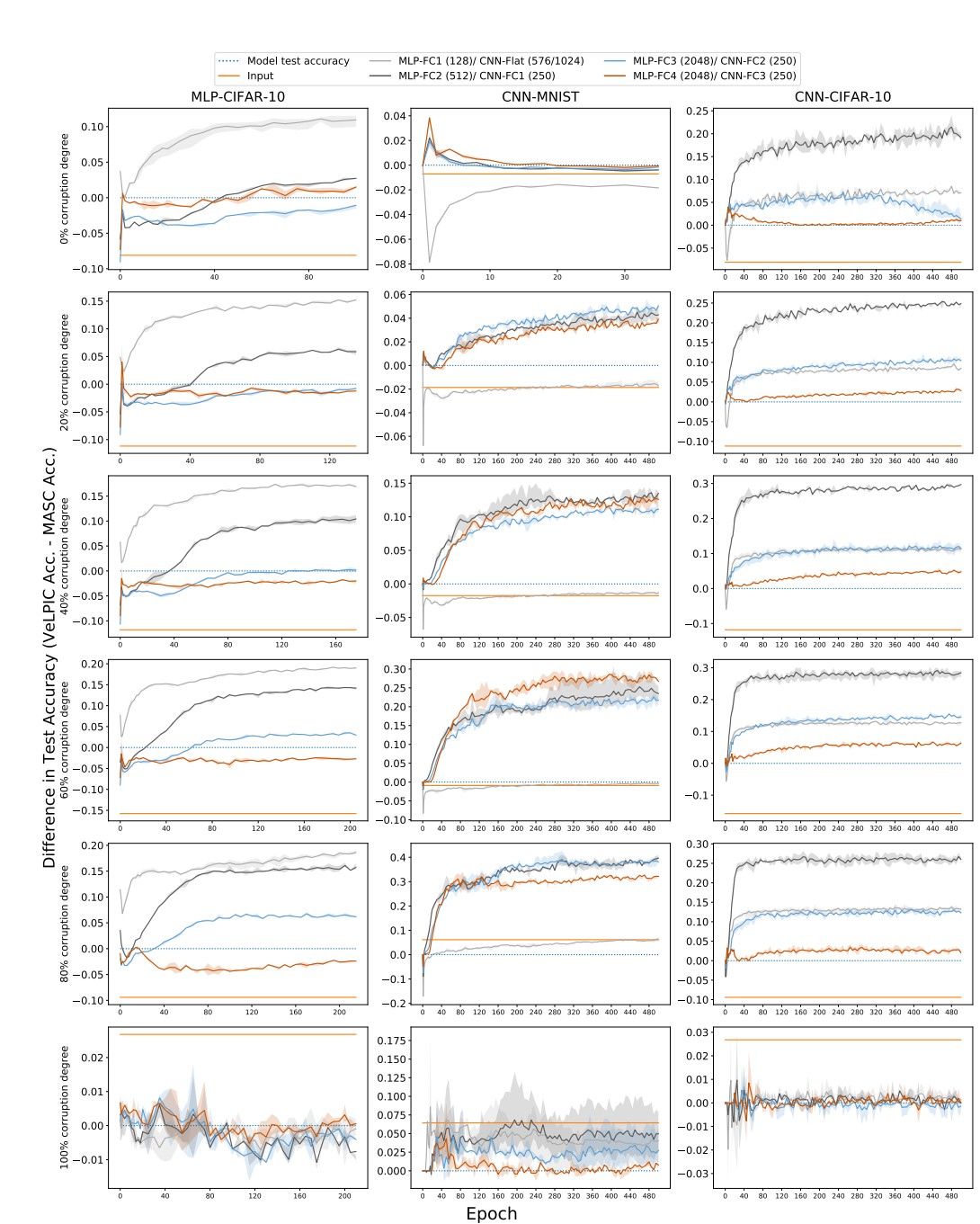

Figure 11: Difference in test accuracy (VeLPIC Accuracy - MASC Accuracy) during training of the network, where test data is projected onto class vectors constructed at each epoch from training data with the indicated label corruption degrees. The plots display difference in accuracy across different layers of the network for various model–dataset combinations. For reference, the test accuracy of the models (blue dotted line) over epochs of training is also shown, which would be 0.

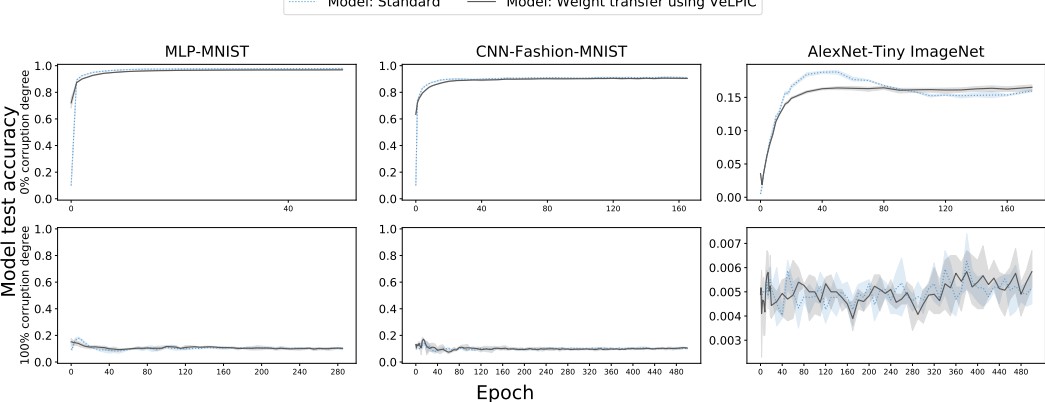

Figure 12: Comparing model test accuracy with VeLPIC transferred accuracy when the weight intervention is applied to the model at the epoch in question during training for corruption degrees 0% and 100%. The test accuracy of the model with standard training without weight intervention (blue dotted line) is overlaid for comparison.

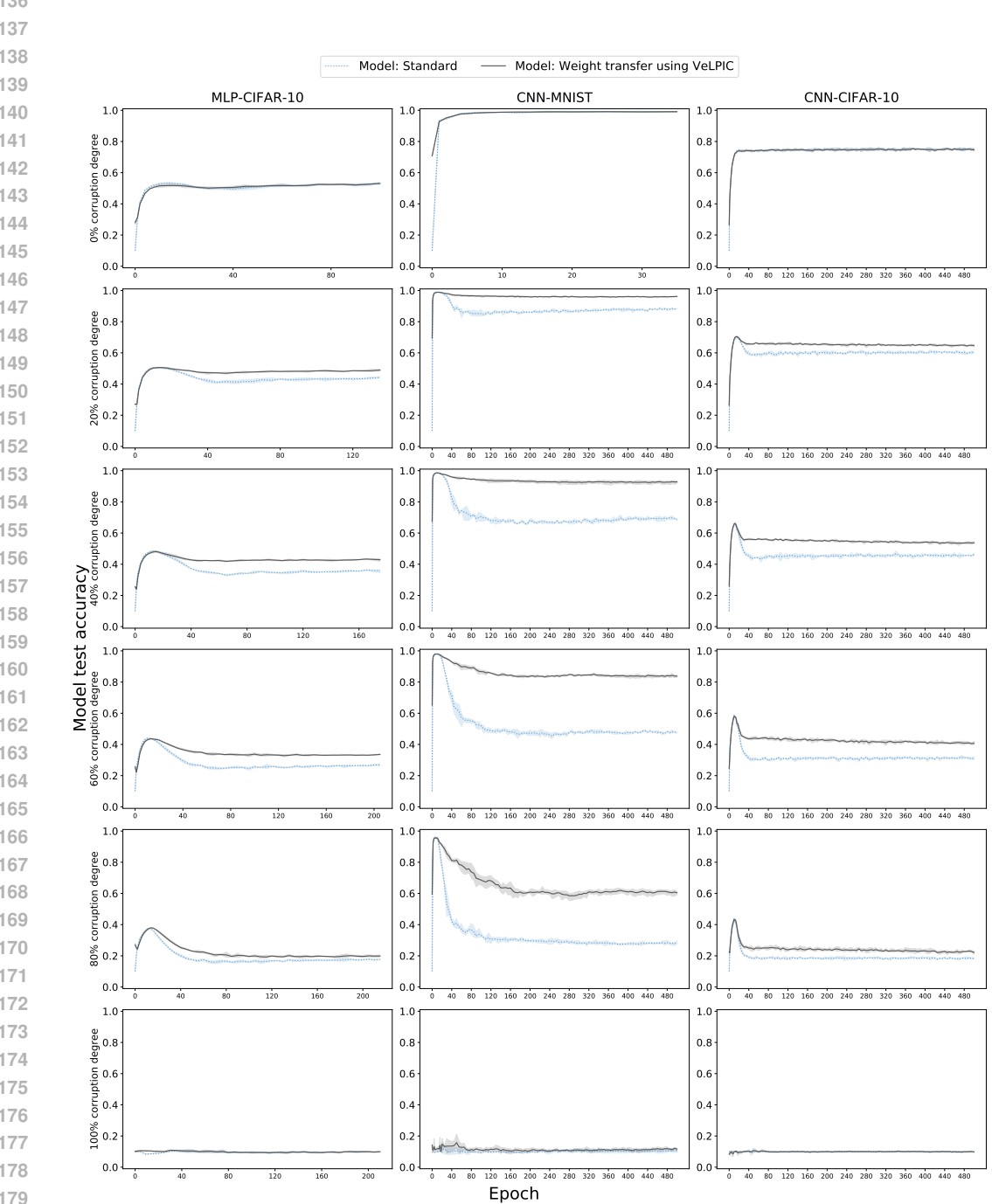

Figure 13: Comparing model test accuracy with VeLPIC transferred accuracy when the weight intervention is applied to the epoch in question during training. The test accuracy of the model with standard training without weight intervention (blue dotted line) is overlaid for comparison.

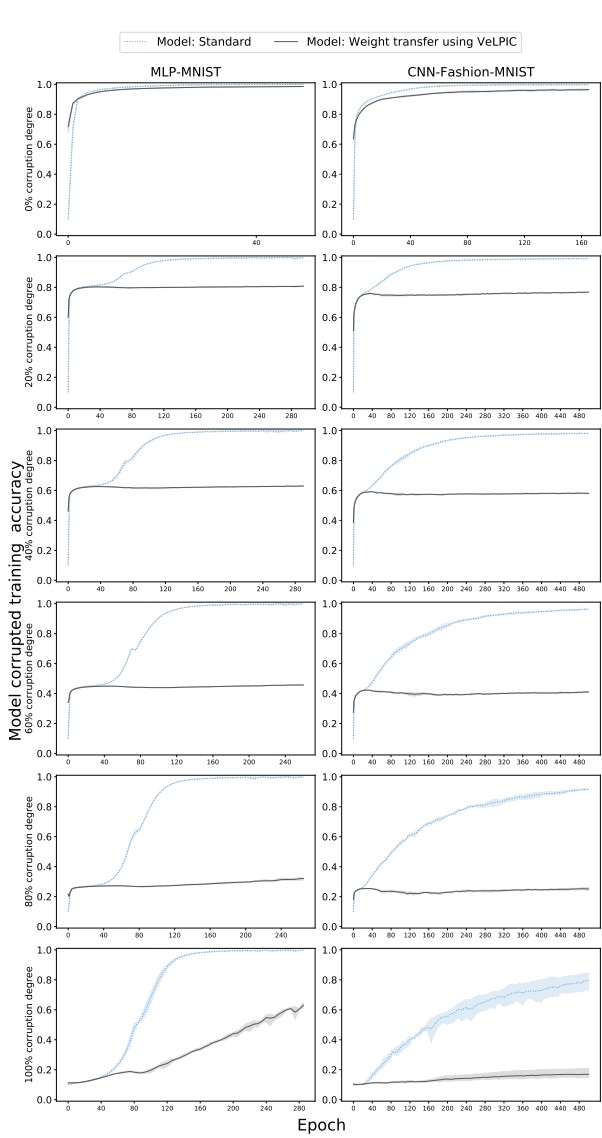

Figure 14: Model train accuracy on corrupted dataset when the VeLPIC weight intervention is applied to pre-softmax weights at the epoch in question during training. The training accuracy on corrupted dataset of the model with standard training without weight intervention (blue dotted line) is overlaid for comparison. Observe that, except for 100% corruption degree, the transferred training accuracy tends to saturate at a level largely consistent with the fraction of true training labels in the corrupted dataset.

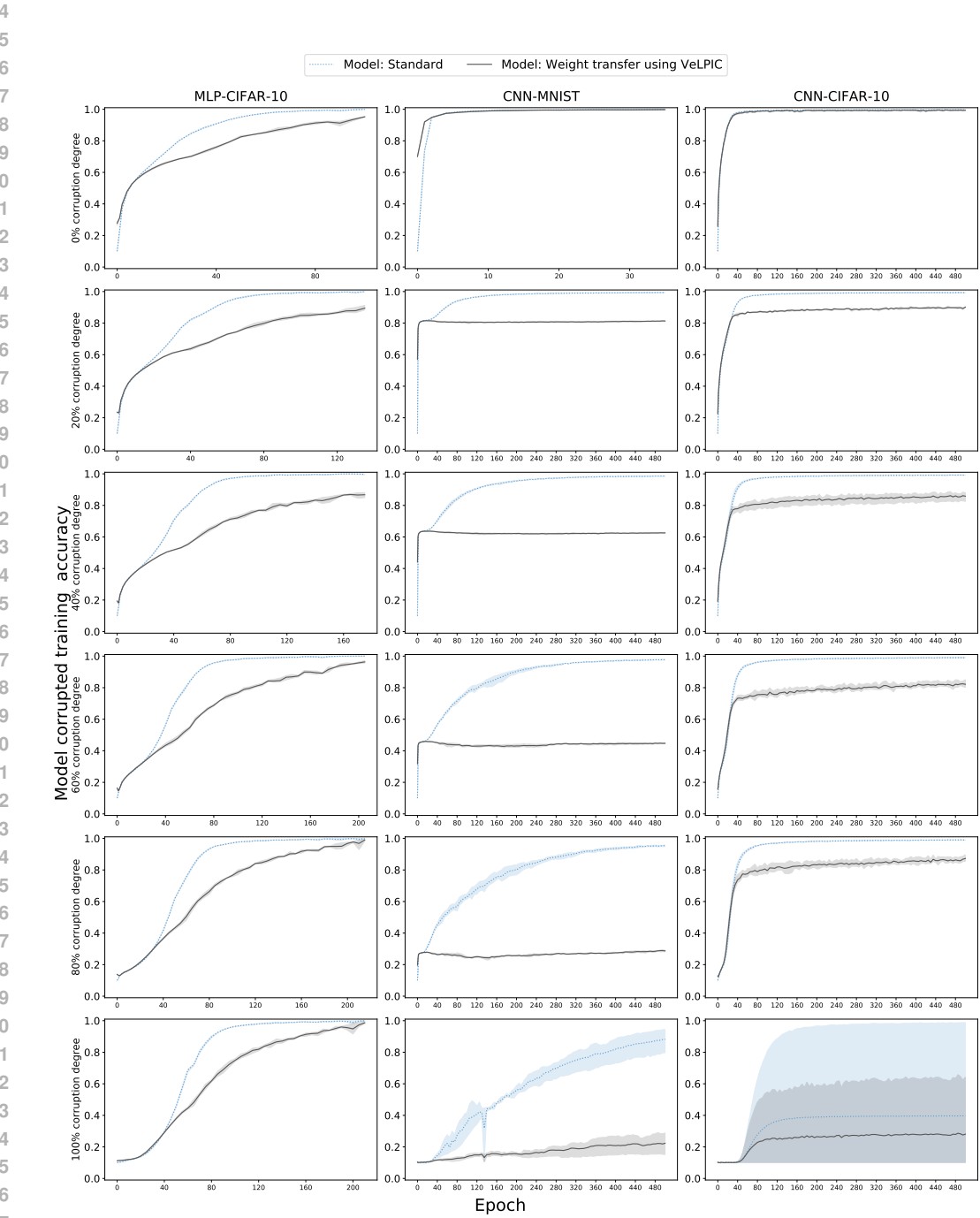

Figure 15: Model train accuracy on corrupted dataset when the VeLPIC weight intervention is applied to pre-softmax weights at the epoch in question during training. The training accuracy on corrupted dataset of the model with standard training without weight intervention (blue dotted line) is overlaid for comparison.

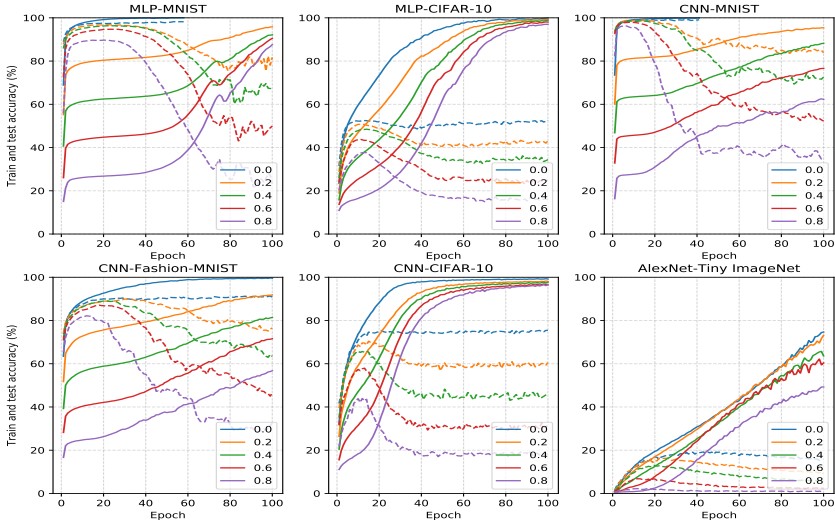

Figure 16: Model's train accuracy with corrupted labels (dash) and test accuracy with true labels (dotted) for different corruptions over the epochs when model is trained on corrupted labels without any intervention. This is consistent with corresponding results reported in (Arpit et al., 2017).

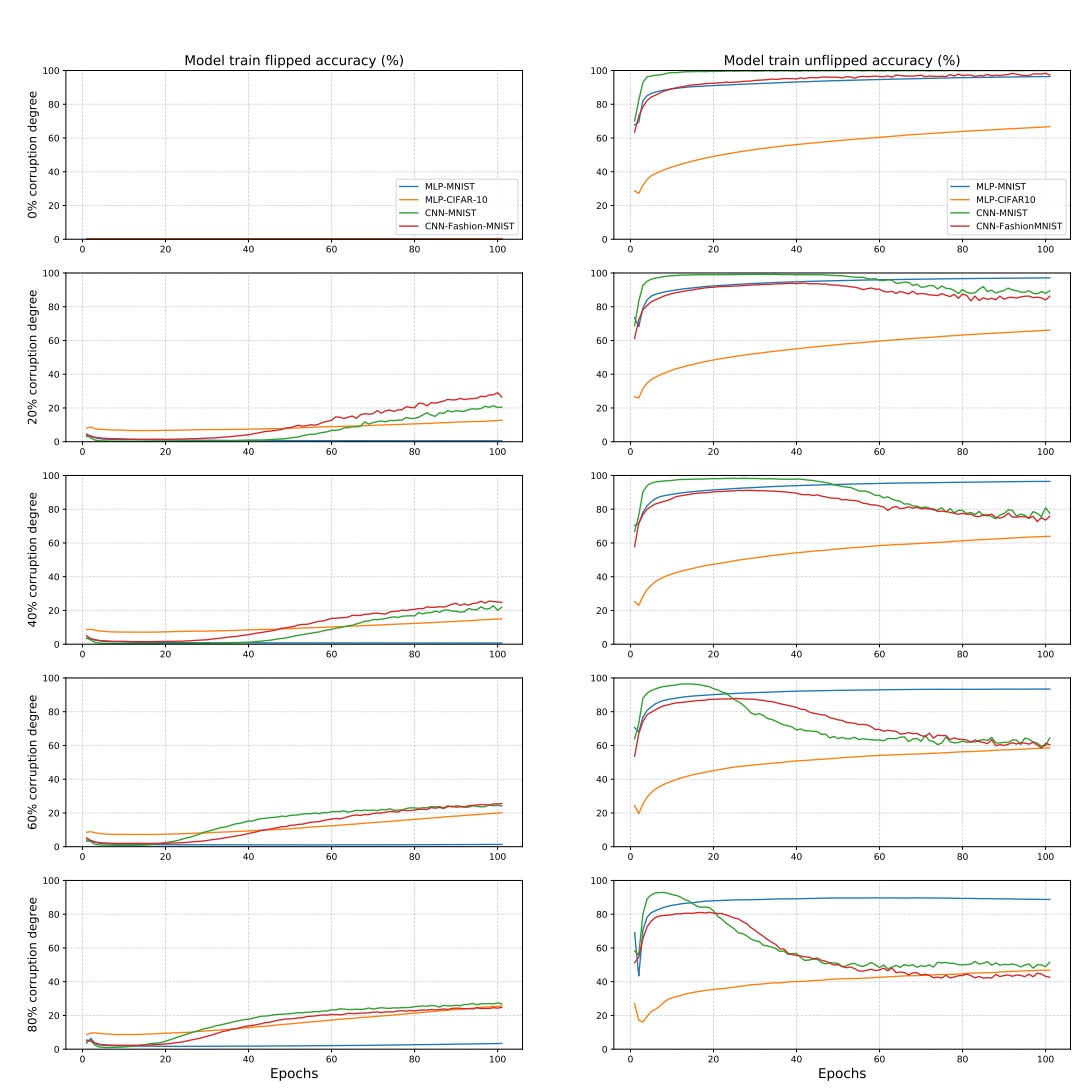

Figure 17: We track separately accuracies on the part of the training data whose labels were changed ("flipped") and unchanged ("unflipped") for the memorization-resistant initializations. Observe that with this initialization, the model training accuracy on flipped labels tends to remain low, whereas model accuracy on unflipped (i.e. true labels) is often quite high. This suggests that the initialization has a tendency to resist memorization. The results are plotted across epochs and for different corruption degrees.

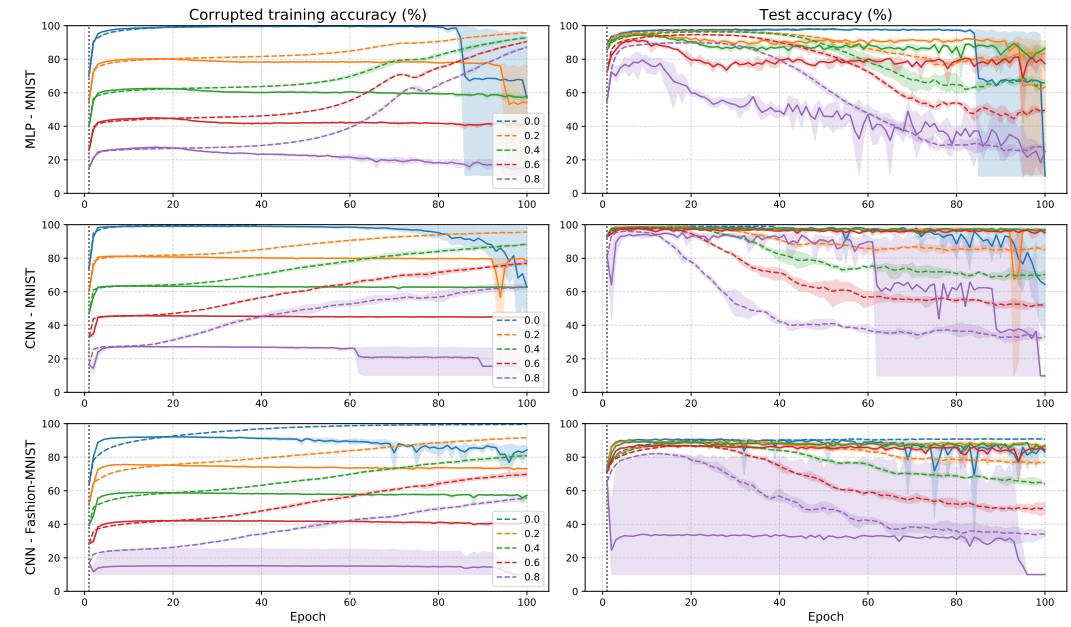

Figure 18: Model train accuracy with corrupted labels and model test accuracy with true labels during training when intervention is performed at random initialization and standard training is performed thereafter. A model with random initialization is loaded. Model weights of the pre-softmax layer were replaced with the VeLPIC class vectors. The model with standard training (dotted) without intervention is overlaid for comparison.

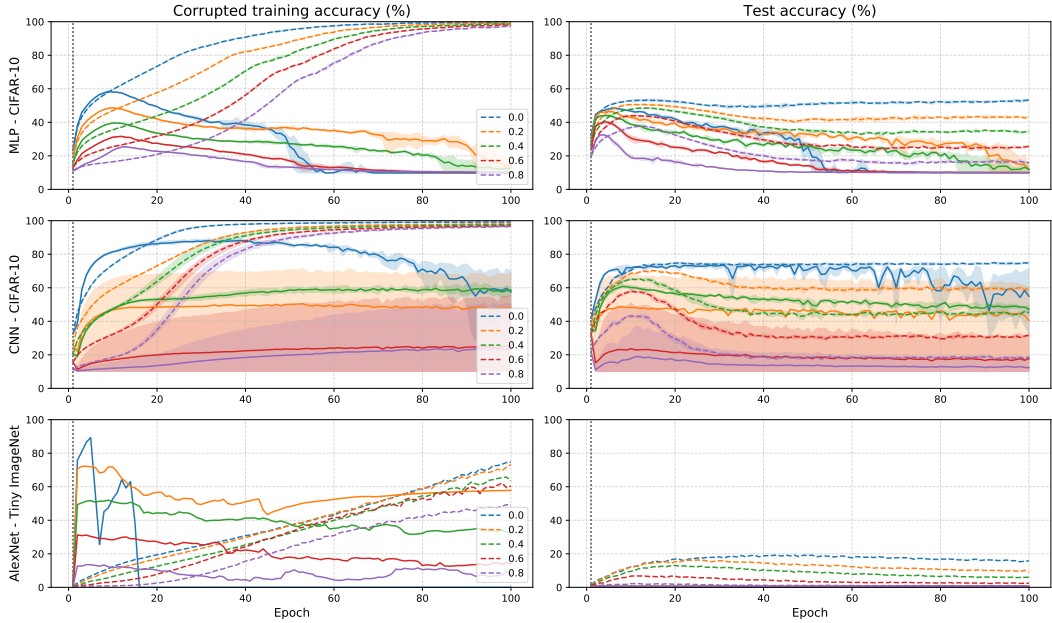

Figure 19: Model train accuracy with corrupted labels and model test accuracy with true labels during training when intervention is performed at random initialization and standard training is performed thereafter. A model with random initialization is loaded. Model weights of the pre-softmax layer were replaced with the VeLPIC class vectors. The model with standard training (dotted) without intervention is overlaid for comparison. The results with AlexNet-Tiny ImageNet are shown with only 1 run.

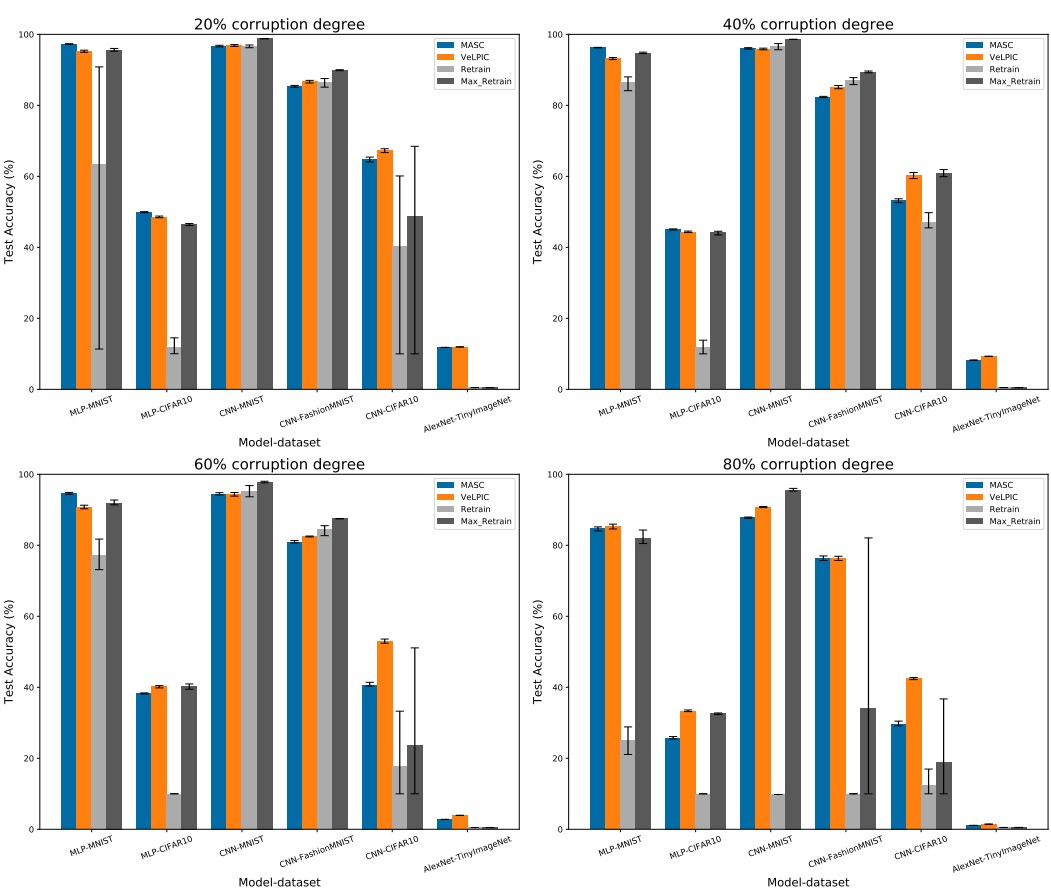

Figure 20: Comparison of model's test accuracy the best-layer MASC test accuracy (with subspaces capturing 99% variance), the best-layer VeLPIC test accuracy, and the test accuracy after applying the intervention at random initialization. For models with the intervention, two values are reported: the test accuracy at the 100th epoch and the maximum test accuracy achieved during training. The error bars indicate the variation observed across three independent runs.

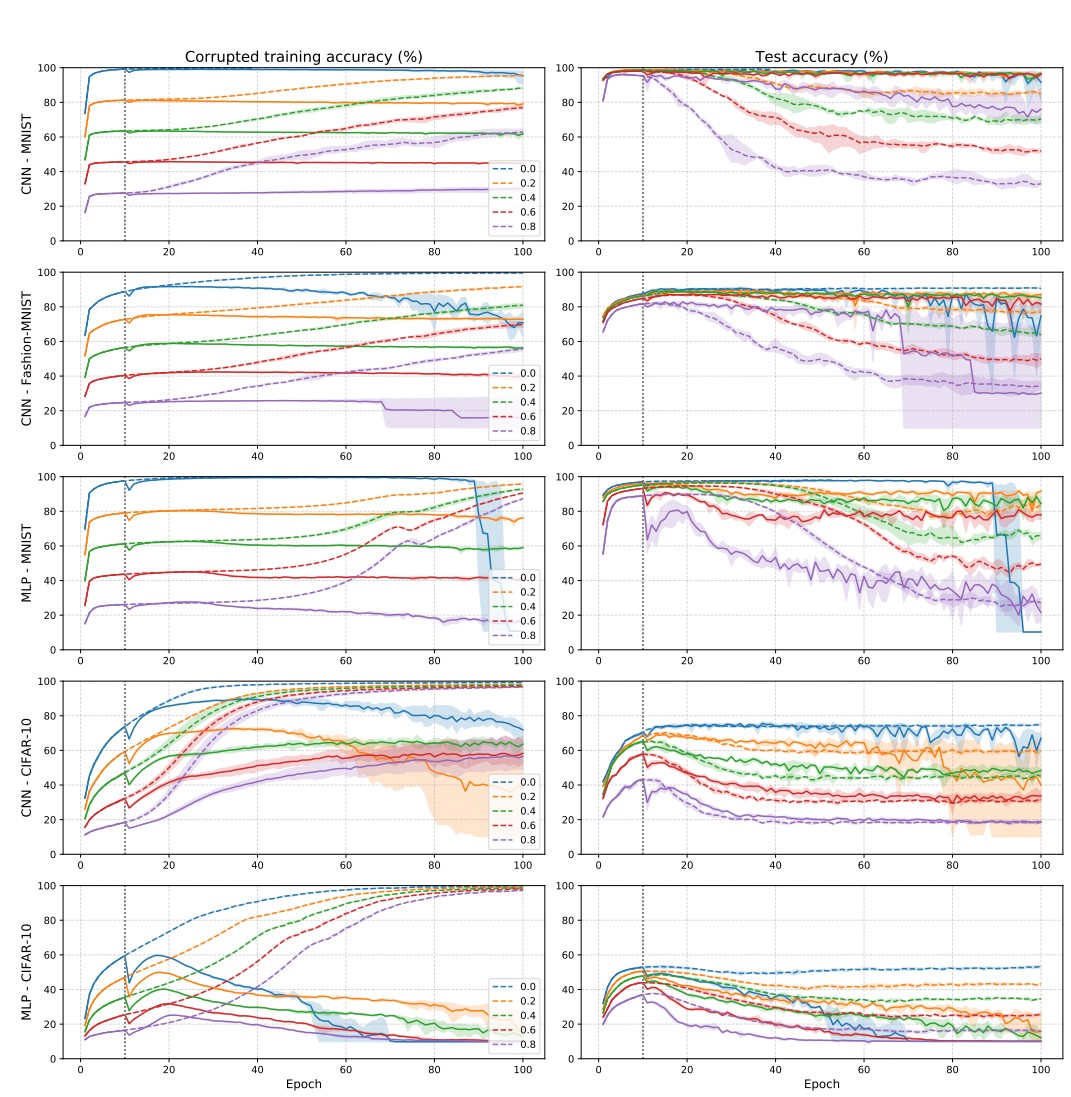

Figure 21: Model train accuracy with corrupted labels and model test accuracy with true labels during training when intervention is performed at 10th epoch and standard training is performed thereafter for 90 epochs. A model is trained using standard training with corrupted dataset is loaded. Model weights of the pre-softmax layer were replaced with the VeLPIC class vectors and trained for 90 epochs. The model with standard training (dotted) without intervention is overlaid for comparison.

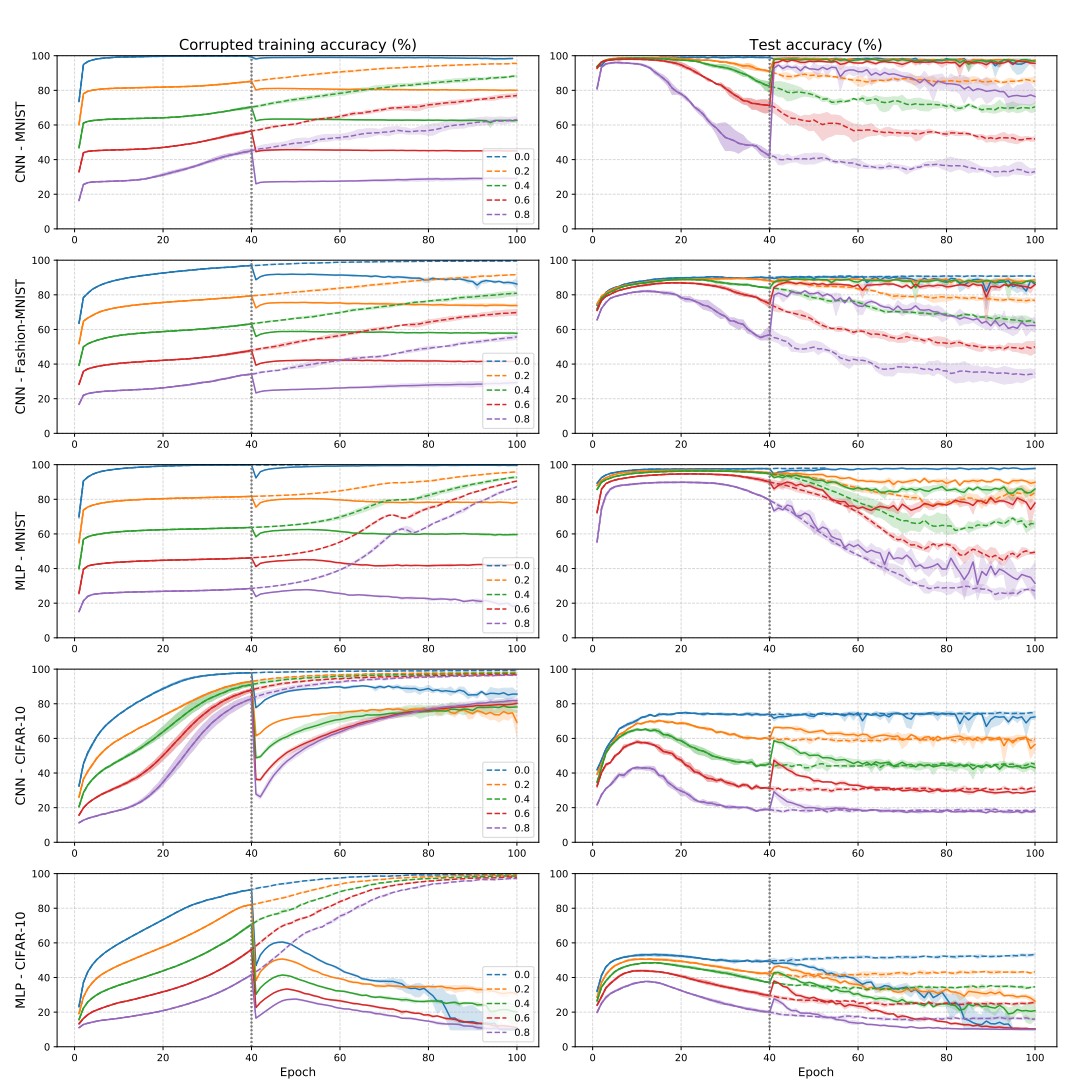

Figure 22: Model train accuracy with corrupted labels and model test accuracy with true labels during training when intervention is performed at 40th epoch and standard training is performed thereafter for 60 epochs. A model is trained using standard training with corrupted dataset is loaded. Model weights of the pre-softmax layer were replaced with the VeLPIC class vectors and trained for 60 epochs. The model with standard training (dotted) without intervention is overlaid for comparison.

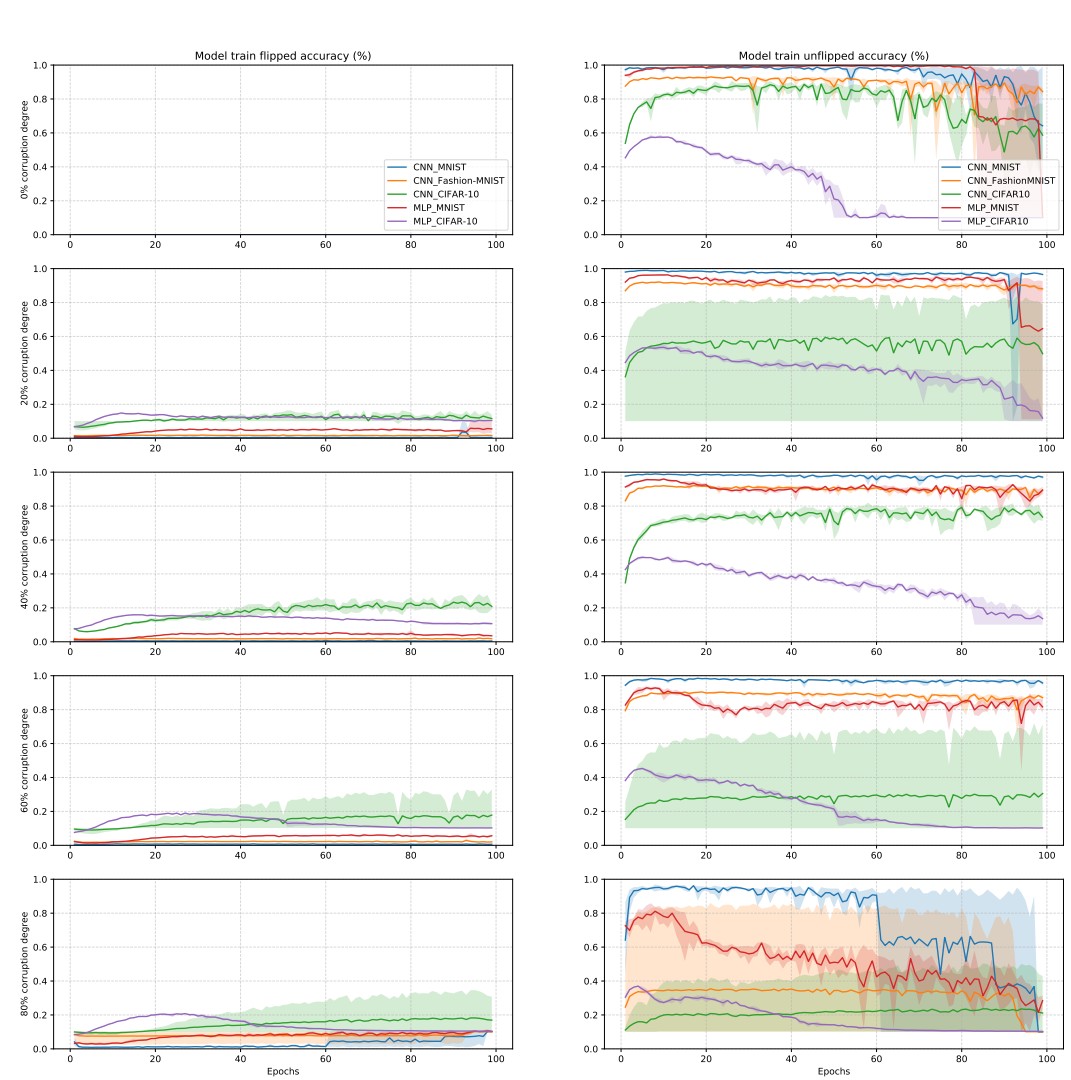

Figure 23: We track separately accuracies on the part of the training data whose labels were changed ("flipped") and unchanged ("unflipped") for the memorization-resistant initializations. Observe that with this initialization, the model training accuracy on flipped labels tends to remain low, whereas model accuracy on unflipped (i.e. true labels) is often quite high. This suggests that the initialization has a tendency to resist memorization. The results are plotted across epochs and for different corruption degrees.

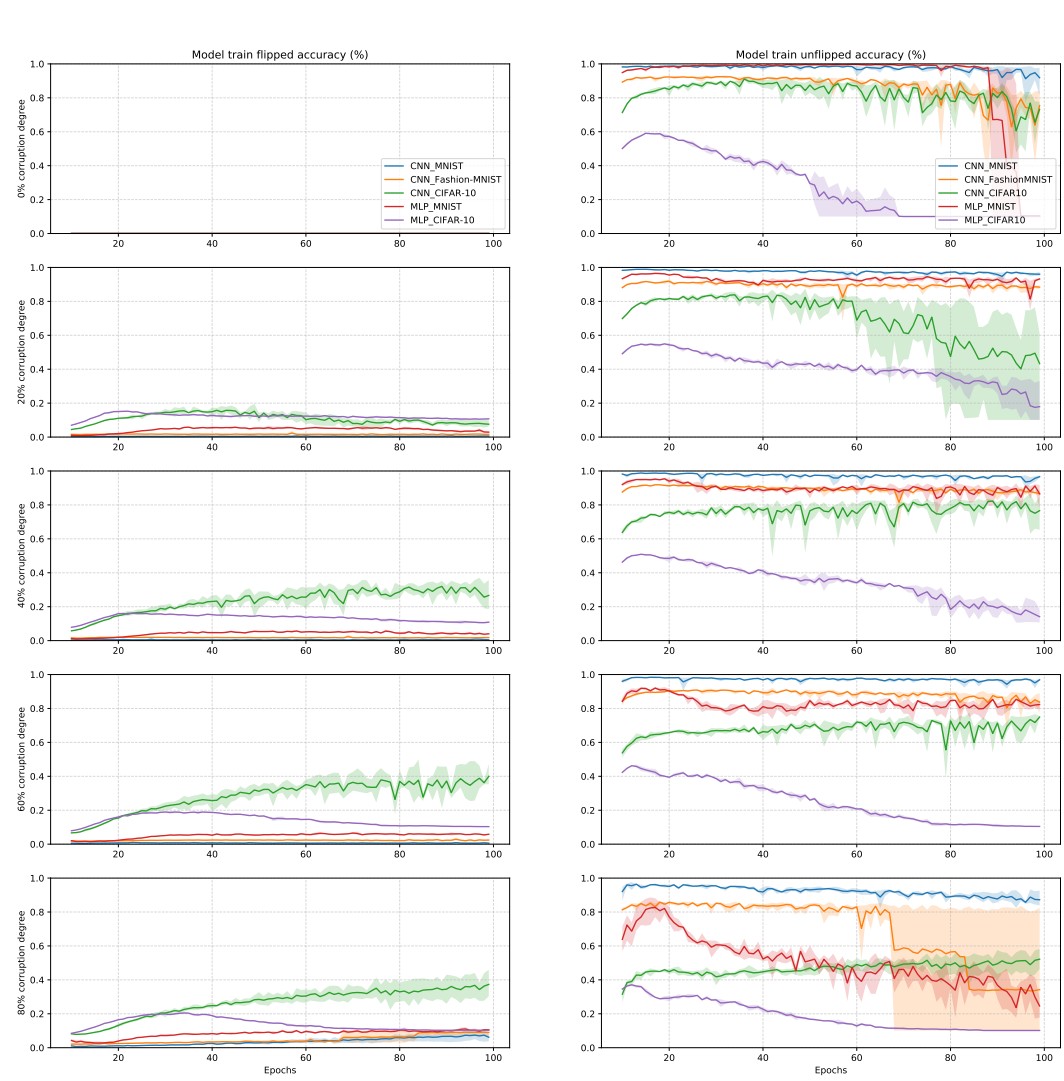

Figure 24: We track separately accuracies on the part of the training data whose labels were changed ("flipped") and unchanged ("unflipped") for the weight initializations performed at 10th epoch. The results are plotted across epochs and for different corruption degrees.

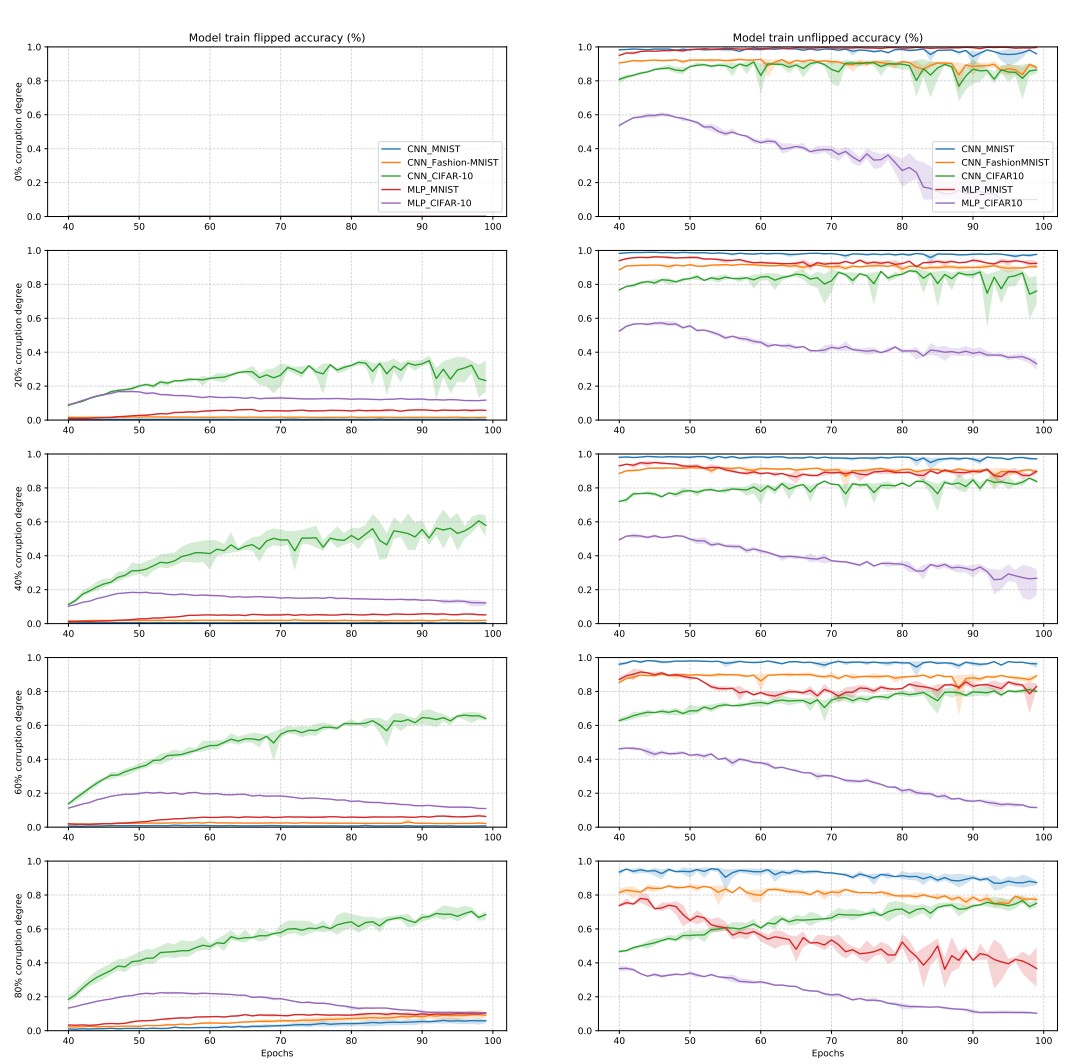

Figure 25: We track separately accuracies on the part of the training data whose labels were changed ("flipped") and unchanged ("unflipped") for the weight initializations performed at 40th epoch. The results are plotted across epochs and for different corruption degrees.

