# OpenReview forum: "On the Dynamics & Transferability of Latent Generalization during Memorization"
_ICLR.cc/2026/Conference — ICLR 2026 Conference Withdrawn Submission_

### Official Review · Reviewer_1JNd · 2025-10-28

**Soundness:** 3
**Presentation:** 2
**Contribution:** 1
**Rating:** 2
**Confidence:** 4

**Summary:**

The paper shows that latent generalization skills persist in models even when they start to memorize corrupted label. The authors propose a new linear probe called Velcip which is then used to identify latent generalization skills in intermediate layers for DNNs. The authors also propose a new initialization scheme which preserves the model's generalization scheme.

**Strengths:**

1) The experiments in the paper are extensive and support the claims made in the paper.
2) The new initialization scheme proposed demonstrates practical utility of the empirical understanding of latent generalization skills.

**Weaknesses:**

My main concern is that the method proposed draws heavily from the prior work [1]. In particular, the variant proposed is just a simple modification of MASC proposed in [1]. While MASC corresponds to using a subspace with the class-specific top-$m$ principal components (where $m$ varies and according to [1] is set as a hyperparameter choice), Velpic is equivalent to using $m=1$. This is because in the first case the quantity being maximized is $(x.p_1)^2$, while Velpic maximizes $x.p_1$, which are equivalent upto a sign (an ambiguity anyways present in PCA). In fact, as the authors notes, the former allows natural sign-invariance, eliminating the need for the mean projection-based sign reversal proposed in the current work.

While it is interesting to note that just using a single component is enough to grant non-trivial improvements in generalization skills, it is necessary to perform a fairer comparison by comparing Velpic with MASC with the top-component (the subspace spanned by $p_1$). This also warrants some more investigations into the statistical properties of the latent representations, eg. whether the top-component already captures a significant portion of the total variance.

Also, alignment preserves accuracy (a discontinuous function) but not loss. The paper does not discuss what happens to the loss at all, restricting the utility of the method to classification tasks only.

[1] Simran Ketha and Venkatakrishnan Ramaswamy. Decoding generalization from memorization in deep neural networks. arXiv preprint arXiv:2501.14687, 2025

**Questions:**

1) Clarification on Fig 2 - is the single curve for the Velpic accuracy the average accuracy across all the layers? Also for completeness the training accuracy should also be plotted.
2) For section 7 (figure 4), the plots end at around epoch 100, as opposed to >200 epochs for the experiments done in Fig. 2. What happens when the model is trained longer? Does the generalization ability persist all throughout?

---

### Official Review · Reviewer_iZq6 · 2025-10-31

**Soundness:** 3
**Presentation:** 2
**Contribution:** 2
**Rating:** 4
**Confidence:** 4

**Summary:**

This paper investigates the dynamics of the generalization power of the learned representation when memorization occurs. It suggests that the generation power in the latent representation persists even under memorization. On top of validating this dynamics using an existing probe (MASC), the paper proposes a new probe (VELPIC) that recovers the generalization ability. Last, it empirically shows the effectiveness of the new probe in variou settings.

**Strengths:**

The paper dives into an interesting problem: the dynamics of learning under potential memorization and what we can still gain from it. It touches quite a few sub-directions in this topic and contributes at various angles.

**Weaknesses:**

My main concern for this work is its lacking of focus. There at least four items of investigation: 1) the dynamics of generalization power in latent representation vs. training epochs, 2) the underlying math nature of an existing probe, 3) a new probe, and 4) how the new probe can be used to transfer the generalization power to another models. Given that all these problem settings are relatively new, the authors may be putting too many good things in one paper: none of the problem settings is introduced or motivated well. See my questions for potential sources of confusion.

The current draft is extremely text-heavy. For some of the background concepts, a good picture can tell a thousand words.

**Questions:**

1) Why do we want to transfer the latent representation to another task/model? What's the common example we have in literature, and does it match the experiment set up? Asking because the experiments are using MNIST, which doesn't really need transferring from larger models for good performance.

2) I assume we have a base model that suffers from certain degree of memorization. Are MASC/VELPIC modifications to the base model that makes it useful for other tasks?

3) Following 2), why is transferability the primary concern? Can MASC/VELPIC be used to salvage the model for its original task?

4) What is the limitation of MASC, or in other words, what motivates us to find a new problem like VELPIC?

---

### Official Review · Reviewer_A95x · 2025-10-31

**Soundness:** 2
**Presentation:** 3
**Contribution:** 1
**Rating:** 2
**Confidence:** 4

**Summary:**

This paper investigates the training dynamics of neural networks when the data is noisy and analyzes the phenomenon that the direct predictions of the model get worse throughout training, while classifiers built on top of the hidden states of the network remain more robust. The authors track the performance of such classifiers and observe that initially, their performance is closely aligned with the original model, but after a certain point, the model starts to perform significantly worse, whereas the classifiers remain more robust.
The authors then develop a new classifier coined Velpic, essentially a linear probe that can outperform previously employed probes of quadratic nature. The new linear probe can directly be incorporated into the networks weights, and also serves as a more robust initialization.

**Strengths:**

1. The paper is well-written and easy to follow. The phenomenon studied is interesting and a better understanding of it would reveal a lot about the training dynamics of neural networks and their surprising robustness to noise.
2. The results on the network being more robust to noise when initialized with the Velpic vectors is interesting. It does depend on the training data used for Velpic, as I'm asking later in this review, but I find it surprising that such an intervention before training can help steer the model away from overfitting.

**Weaknesses:**

1. The contributions of this work are not very clear to me. The work largely builds upon the previous work [1] that introduced the MASC classifier, and the initial set of experiments seem to largely obtain the same insights. What seems maybe novel is the dynamics aspect by tracking the performance throughout training but the behavior is not very surprising to me.
The new classifier, Velpic, seems like a more complicated way of training a linear classifier on top of the hidden states instead of just using the cross entropy? What I however did not understand is the training data for it: are you using the same noisy data or do you assume access to the ground truth labels?
Using linear probes on top of neural networks has by now a very long tradition in self-supervised learning and other areas, while Velpic seems to use another algorithm to find the linear classifier, I highly doubt that results would vary much if cross entropy were used.
2. I'm also confused by the section on weight transfer to the original model, where essentially the head of the model is replaced by the Velpic vectors. Given that this is a linear classifier, isn't it by definition going to be the exact same predictions? I'm not sure if I'm missing something but this seems very obvious and not worth writing an entire section for. Even the plots in Figure 2 and Figure 3 look exactly identical, I don't think it is worth repeating the exact same plot for this observation.

[1] Ketha et al., Decoding generalization from memorization in deep neural networks, 2025

**Questions:**

1. Do you use data augmentation during training? Previous work has observed very strong generalization even under perfect label noise when training with heavy data augmentation and subsequently training a linear classifier with the clean data [1]. Some of your observations might relate to those results, or might be amplified if data augmentation is used.
2. According to the plots regarding robustness to noise when initializing with Velpic weights, it seems that MLPs benefit a lot but CNNs way less, it seems to even hurt them to some degree according to Figure 4. Do you have an intuition on this discrepancy?

[1] Anagnostidis et al, The curious case of benign memorization, 2022

---

### Official Review · Reviewer_W8Qc · 2025-11-01

**Soundness:** 1
**Presentation:** 1
**Contribution:** 1
**Rating:** 2
**Confidence:** 4

**Summary:**

The paper tracks how “latent generalization” emerges and persists in networks trained to memorize corrupted labels, argues MASC is a quadratic probe, proposes a new linear probe (VeLPIC) that often scores higher, and claims one can transfer the latent signal into the model’s weights and craft “memorization-resistant” initializations.

**Strengths:**

* This paper tries to address a timely problem in machine learning

**Weaknesses:**

* Latent generalization is not well defined, no evidence for this is phenomena provided (or well cited reference).
* The MASC-is-quadratic proof feels orthogonal to the central empirical story; the paper doesn’t explain why that nonlinearity matters for practice or theory
* Evaluation is limited to small datasets/older architectures; there’s no evidence that claims hold on modern large-scale setups (e.g., ResNets/ViTs on ImageNet-1k, language models).
* Writing/organization are rough, making it hard to follow; definitions and motivation need tightening.

**Questions:**

1. What is MASC?
2. How is it trained?
3. Why does MASC help and not linear probe?
4. Why do we care that MASC is quadratic?
5. Can you provide evidence for latent generalization on large scale image/language datasets?

---

> ### Author Response · Authors · 2025-11-20
>
> We are disappointed that the reviewer failed to notice that many of the questions/comments raised have already been addressed quite prominently in the paper.
>
> We list some of them below and point out passages in the paper, where they have been directly addressed.
>
> * *"Latent generalization is not well defined, no evidence for this is phenomena provided (or well cited reference)."*
>
> Lines 50-53 of the submitted paper:
> > A recent study (Ketha & Ramaswamy, 2025) has shown that while Deep Networks trained on
> datasets having corrupted labels tend to exhibit poor generalization, their intermediate layer representations retain a surprising degree of latent generalization ability. This ability can be recovered
> from such trained networks by using a simple probe – Minimum Angle Subspace Classifier (MASC)
> – that leverages the subspace geometry of the corrupted training dataset representations, to this end.
>
> * *"The MASC-is-quadratic proof feels orthogonal to the central empirical story; the paper doesn’t explain why that nonlinearity matters for practice or theory"*
>
>     *"Why do we care that MASC is quadratic?"*
>
> Lines 219-224 of the submitted paper:
> > Given that MASC is inherently a non-linear classifier as proved above, a natural question is if its
> extraordinary ability to decode generalization from hidden representations of memorized networks
> is a consequence of its non-linearity. Put differently, it raises the question of whether the latent
> generalization reported in (Ketha & Ramaswamy, 2025) is linearly decodable – with comparable
> performance – from the layerwise representations of the network.
>
> * *"1. What is MASC?" "2. How is it trained?"*
>
> Lines 130-139 of the submitted paper:
> > (Ketha & Ramaswamy, 2025) investigate the organization of class-conditional subspaces using the
> training data at various layers of Deep Networks.
> These subspaces are estimated via Principal Components Analysis (PCA), specifically, ensuring that
> they pass through the origin. To probe the layerwise geometry without relying on subsequent layers,
> they propose a new probe – the Minimum Angle Subspace Classifier (MASC). For a given test input,
> MASC projects the layer output onto each class-specific subspace, and computes the angles between
> the original and projected vectors, for each subspace. The label predicted by MASC corresponds
> to the class whose subspace yields the projected vector with the smallest such angle. We provide a
> detailed summary of the working of MASC in the Appendix Section A.2.

---

### Note · Authors · 2025-11-20

**Comment:**

We thank the reviewers for their time and feedback.

We are withdrawing the paper in order to allow us the time to do a comprehensive revision, which we will submit elsewhere.  Although, we disagree with some of the points raised, in many cases, better exposition will serve to motivate some of these questions better; we are also considering additional experiments in a few cases and planning to overhaul the writing.

**Withdrawal Confirmation:**

I have read and agree with the venue's withdrawal policy on behalf of myself and my co-authors.